# CONSTRAINED GRAPH CLUSTERING WITH SIGNED LAPLACIANS

## ABSTRACT

Given two weighted graphs $G = (V, E, w_G)$ and $H = (V, F, w_H)$ defined on the same vertex set, the constrained clustering problem asks to find a set $S \subset V$ that minimises the cut ratio between $w_G(S, V \setminus S)$ and $w_H(S, V \setminus S)$. We develop a Cheeger-type inequality that relates the solution of the constrained clustering problem to the spectral properties of $G$ and $H$. To reduce computational complexity, we use the signed Laplacian on $H$, simplifying the calculations while maintaining accurate results. By solving a generalized eigenvalue problem, our algorithm provides improvements in performance, particularly in scenarios where traditional spectral clustering methods face difficulties. We demonstrate its practical effectiveness through experiments on both synthetic and real-world datasets.

## 1 INTRODUCTION

Clustering is a fundamental technique in machine learning, with extensive applications across computer science and various scientific disciplines. The primary goal of clustering is to partition data points into clusters, such that points within each cluster are more densely connected than those in other clusters. Traditional clustering, such spectral clustering (Von Luxburg, 2007) rely solely in the structure of the data. However, in many real-world scenarios, additional domain knowledge is available, introducing specific constraints that should be incorporated into the clustering process to achieve more accurate and meaningful results (Basu et al., 2008).

Constrained clustering focuses on developing algorithms that effectively incorporate this domain knowledge to enhance clustering performance (Wagstaff et al., 2001). The domain knowledge is represented by two types of pairwise constraints: (1) MUST-LINK constraints, requiring that a pair of data points be assigned to the same cluster, and (2) CANNOT-LINK constraints, requiring that a pair of data points must be assigned to different clusters. In the context of graph clustering, the goal is partition the vertices of a graph based on edge connectivity while satisfying the given constrains.

In this paper, we examine the constrained clustering problem by formalize these constraints using the graphs $G = (V, E, w)$ and $H = (V, E', w')$, in which every data point corresponds to a graph vertex, every MUST-LINK (resp, CANNOT-LINK) constraint corresponds to an edge in $G$ (resp, $H$), and the edge weights capture the strength of the user's preference for satisfying the corresponding constraint. For any set $S \subseteq V$, we define the cut ratio of $S$ between $G$ and $H$ by

$$\text{cut}_H^G(S, V \setminus S) = \frac{w^G(S, V \setminus S)}{w^H(S, V \setminus S)}, \tag{1}$$

and the objective is to find $S$ that achieves

$$\Phi_H^G = \min_{\emptyset \subset S \subset V} \text{cut}_H^G(S, V \setminus S).$$

We develop an efficient approximation algorithm for the constrained graph clustering problem. The key to our algorithm is a Cheeger-type inequality that upper bounds $\Phi_H^G$ with respect to $\lambda_2(\Delta_H^G)$ and $\lambda_2(\Delta^H)$, where

$$\lambda_2\left(\Delta_H^G\right) = \min_{x \perp \mathbf{1}} \frac{\langle x, \Delta^G x \rangle}{\langle x, \Delta^H x \rangle}, \quad \text{and} \quad \lambda_2\left(\Delta^H\right) = \min_{x \perp \mathbf{1}} \frac{\langle x, \Delta^H x \rangle}{\langle x, x \rangle}. \tag{2}$$

and $\Delta^G$ and $\Delta^H$ are the Laplacian operators of $G$ and $H$ respectively. By introducing several techniques to adjust $G$ and $H$, we significantly reduce the time complexity for solving the generalised

eigenvalue problem, while maintaining the one-to-one correspondence between the solution of the new reduced instance and the initial one. The empirical studies on both the synthetic and real-world data sets confirm that with the two sets of constraints our algorithm presents significantly better performance than the classical spectral clustering algorithm, and the running time of our algorithm is close to traditional spectral clustering methods.

**Related work.** Cheer-type inequalities for constrained graph clustering are studied in literature. For example, Cucuringu et al. (2016) proved that $\Phi_H^G \cdot \Phi_K^G \leq 4\lambda_2(\Delta_H^G)$; this inequality is based on a third graph $K$, which they call the demand graph. Koutis et al. (2023) showed that $\Phi_H^G \leq 16\lambda_2(\Delta_H^G)/\Phi(G)$, where $\Phi(G)$ is the standard conductance of $G$. These two result cannot be directly compared with ours, since both inequalities upper bound $\Phi_H^G$ with respect to $\lambda_2(\Delta_H^G)$ and parameters of $H$, i.e., $\Phi_K^G$ in (Cucuringu et al., 2016) and $\Phi(G)$ in (Koutis et al., 2023). In contrast, we upper bound $\Phi_H^G$ with respect to $\lambda_2(\Delta_H^G)$ and $\lambda_2(\Delta^H)$. Trevisan (2013) studied the computational complexity of the problem, and proved that under the Unique Games Conjectures it's impossible to find a cut that achieves $O\left(\sqrt{\Phi_H^G}\right)$ approximation in polynomial time.

Our work also relates to the studies on constrained graph clustering from practical perspectives, e.g., (Jia et al., 2021; Wang & Davidson, 2010; Wang et al., 2014) and signed cuts using the signed Laplacian (Knyazev, 2017). Most of these studies, however, lack a rigorous analysis on the quality of the resulting clusters compared to the optimal solution. Our work is further linked to Cheeger-type inequalities for different graph Laplacians (Lange et al., 2015; Li et al., 2019) (including the signed Laplacian (Atay & Liu, 2020)), and their higher-order generalisations (Lee et al., 2014).

**Contribution.** We present a constrained graph clustering method and establish a novel Cheeger-type inequality that directly relates the problem to the spectral properties of the graphs $G$ and $H$, providing theoretical guarantees often missing in existing approaches. We also introduce an efficient implementation leveraging the signed Laplacian, which simplifies computations and ensures stability without requiring additional parameters. Finally, we demonstrate the robustness and efficiency of our algorithm on both synthetic and real-world datasets.

## 2 BACKGROUND & PRELIMINARIES

We consider a finite, undirected graph $G = (V, E)$, where $V$ is the set of vertices and $E$ is the set of edges. Each edge $uv \in E$ denotes an undirected connection between vertices $u$ and $v$. A self-loop in this context is represented by $uu$, indicating an edge that starts and ends at the same vertex $u$. The notation $u \sim v$ means that $u$ and $v$ are connected by an edge. We define weights in the graph $G$ through the function $w : E \to \mathbb{R}^+$, where $w_{uv} = w_{vu}$ specifies the weight of the edge between vertices $u$ and $v$. By definition, each self-loop $uu$ contributes twice to the degree of vertex $u$.

For $E_0 \subseteq E$, we interpret $w$ as a *discrete measure* on the corresponding sets, using the notation $w(E_0) = \sum_{uv \in E_0} w_{uv}$. For $V_0, V_1 \subseteq V$, we denote the set of unoriented edges between $V_0$ and $V_1$ as $E(V_0, V_1) = \{uv \in E \mid u \in V_0 \text{ and } v \in V_1\}$. We denote by $E_v$ the set of all edges connected to $v$, and $N_v$ the neighbourhood of $v$ as the set of vertices adjacent to $v$, i.e., $E_v = E(\{v\}, V)$ and $N_v := \{u \in V \mid v \sim u\}$. The edge weights on a graph determine a *weighted degree* of a vertex, defined by $\deg(v) = w(E_v) = \sum_{e \in E_v} w_e$.

**The Laplacian.** We define some standard spaces related to a finite and weighted graph $G = (V, E)$ with weight function $w$. We define the Hilbert spaces $\ell_2(V, w)$ and $\ell_2(E, w)$ as:

$$\ell_2(V, w) = \{\varphi : V \to \mathbb{R}\}, \quad \ell_2(E, w) = \{\eta : E \to \mathbb{R}\}.$$

Furthermore, we consider the natural inner product for these spaces. For $\ell_2(V, w)$, the inner product between functions $\varphi$ and $\psi$ is defined as $\langle \varphi, \psi \rangle_V = \sum_{v \in V} \varphi(v)\psi(v)\deg(v)$. For $\ell_2(E, w)$, the inner product between functions $\eta$ and $\xi$ is defined as $\langle \eta, \xi \rangle_E := \sum_{e \in E} \eta_e \xi_e w_e$. Let $G = (V, W, w)$ be a weighted graph. The *derivative* $d$ is defined as

$$d : \ell_2(V, w) \longrightarrow \ell_2(E, w), \quad (d\varphi)_{e=(u,v)} = \varphi(u) - \varphi(v).$$

The adjoint $d^* : \ell_2(E, w) \longrightarrow \ell_2(V, w)$ is given by

$$(d^*\eta)(v) = -\frac{1}{\deg(v)} \sum_{e \in E_v} w_e \eta_e.$$

The weighted Laplacian $\Delta : \ell_2(V, w) \longrightarrow \ell_2(V, w)$ is defined as $\Delta = d^*d$, and acts as

$$(\Delta\varphi)(v) = \frac{1}{\deg(v)} \sum_{u \in N_v} \big(\varphi(v) - \varphi(u)\big) w_{uv}.$$

Let $G = (V, E, w)$ be a weighted graph, and for all $\varphi \in \ell_2(V, w)$ we have that

$$\langle \varphi, \Delta^G \varphi \rangle_{\ell_2(V)} = \langle \varphi, d^*d\varphi \rangle_{\ell_2(V)} = \langle d\varphi, d\varphi \rangle_{\ell_2(E)} = \sum_{u \sim v} |d\varphi_{uv}|^2 w_{uv} = \sum_{u \sim v} (\varphi(u) - \varphi(v))^2 w_{uv}.$$

**Graph Signature.** A *signature* of a graph $G = (V, E)$ is a map $\alpha : E \to \{+1, -1\}$, which assigns a sign to each edge. Let $G = (V, E, w)$ be a weighted graph with a signature $\alpha$. The *signed Laplacian*, denoted as $\Delta_\alpha$, is a linear operator $\Delta_\alpha : \ell^2(V, w) \to \ell^2(V, w)$, defined by

$$(\Delta_\alpha \varphi)(v) = \frac{1}{\deg(v)} \sum_{u \in N_v} \big(\varphi(v) - \alpha_{vu}\varphi(u)\big) w_{vu},$$

where $w_{uv}$ is the weight of the edge $uv$, and $\alpha_{uv}$ indicates the sign of the edge as given by the signature $\alpha$. Observe that the Laplacian is a particular case of the signed Laplacian by taking $\alpha_{uv} = 1$ for all edges and the signless Laplacian by taking $\alpha_{uv} = -1$ for all edges. The signed Laplacian can be viewed as a special case of the magnetic Laplacian with a discrete magnetic potential taking values in $\{0, \pi\}$. $\Delta_\alpha$ (and therefore $\Delta$) are positive semi-definite and self-adjoint operators, hence all the eigenvalues are real and non-negative.

# 3 ALGORITHM & ANALYSIS

In this section, we present a constrained graph clustering algorithm called CC++. At a high level, our algorithm consists of the following: in the preprocessing step, we adjust the edge weights of $G$ and add self-loops to the vertices of $G$, such that both of $G$ and $H$ have the same degree sequence. Then, we show that a desired cut can be found by a sweep-set algorithm when the eigenvector corresponding to a generalized eigenvalue problem is given as input. Taking the practical implementation into account, we introduce a negative self-loop in $H$ for efficient computation, and justify its performance both in theory and in practice. Due to page limitations, proofs omitted from this section can be found in the appendix.

## 3.1 PREPROCESSING $G$ AND $H$

In the preprocessing step, our algorithm first scales the weights of all the edges in $G$ by the same factor, and adds self-loops to the resulting graph $G$. These two operations ensure that the constructed graph $\widetilde{G}$ and $H$ have the same degree sequence, while maintaining the optimal cut of the input instance.

**Scaling the graph $G$.** For any $c \in \mathbb{R}^+$, we scale the edge weights of $G$ by a factor of $c$ and define $G(c) = (V, E, c \cdot w)$, where $(c \cdot w)_e = c \cdot w_e$ for each edge $e$. By equation 1, we have that

$$\text{cut}_H^{G(c)}(S, V \setminus S) = \frac{w^{G(c)}(S, V \setminus S)}{w^H(S, V \setminus S)} = c \cdot \frac{w^G(S, V \setminus S)}{w^H(S, V \setminus S)} = c \cdot \text{cut}_H^G(S, V \setminus S).$$

Thus, the minimum cut problem for the scaled graph can be expressed as

$$\Phi_H^{G(c)} = c \cdot \Phi_H^G.$$

This equivalence indicates that the choosing an appropriate scaling factor $c$ is crucial for balancing the edge weights between $G$ and $H$. To ensure that the degrees of the vertices in the scaled graph $G(c_0)$ do not exceed those in $H$, we define the scaling factor $c_0$ by

$$c_0 = \min_{v \in V} \left\{ \frac{\deg^H(v)}{\deg^G(v)} \right\}. \tag{3}$$

This choice of $c_0$ guarantees that for the scaled graph $G(c_0)$, the degree of each vertex $v \in V$ satisfies that $\deg^{G(c_0)}(v) \leq \deg^H(v)$ for all $v \in V$. With this scaling factor $c_0$ established, we now proceed to study the properties of the graph $G = (V, E, w)$ and its corresponding minimum cut value $\Phi_H^G$ in comparison to $H = (V, E', w')$, under the assumption that $\deg^G(v) \leq \deg^H(v)$ for all $v \in V$.

**Equalizing the Degrees of $G$.** To further refine this comparison, consider the subset of vertices

$$V_0 = \left\{ v \in V \mid \deg^G(v) < \deg^H(v) \right\}.$$

We now construct a new graph $\widetilde{G} = (V, \widetilde{E}, \widetilde{w})$, where $\widetilde{E} = E \cup \{(v,v)\}_{v \in V_0}$, and the weight function $\widetilde{w}$ is defined by

$$\widetilde{w}\mid_E = w \quad \text{and} \quad \widetilde{w}(v,v) = \frac{\deg^H(v) - \deg^G(v)}{2} \quad \forall v \in V_0.$$

Observe that the construction ensures that $\deg^{\widetilde{G}}(v) = \deg^H(v)$ for all $v \in V$. Indeed, we have that

$$\deg^{\widetilde{G}}(v) = \sum_{u:\, u \sim v} \widetilde{w}_{uv} = \sum_{u:\, u \sim v} w_{uv} + 2 \sum_{(v,v)} \widetilde{w}(v,v) = \deg^G(v) + (\deg^H(v) - \deg^G(v)) = \deg^H(v).$$

This modification of $G$ demonstrates that, despite the additional restriction imposed by $c_0$, the problems remain equivalent. Specifically, we observe that

$$\Phi_H^G = \min_{S \subseteq V} \frac{w^G(S, V \setminus S)}{w^H(S, V \setminus S)} = \min_{S \subseteq V} \frac{\widetilde{w}(S, V \setminus S)}{\widetilde{w}_H(S, V \setminus S)} = \Phi_H^{\widetilde{G}}.$$

Therefore, the generalized cut problem for $G$ and $H$ remains equivalent to that of $\widetilde{G}$ and $H$, despite the degree adjustments made in $\widetilde{G}$. The following remark will be used in our analysis.

**Remark 1.** *For the weighted graph $G = (V, E, w)$, and the previous $\widetilde{G} = (V, \widetilde{E}, \widetilde{w})$ where $\widetilde{E}$ includes additional self-loops at some vertices, and any function $\varphi : V(G) = V(\widetilde{G}) \to \mathbb{R}$, the following holds: Adding self-loops does not affect the quadratic form associated with the graph Laplacian, i.e., $\langle \varphi, \Delta^G \varphi \rangle = \langle \varphi, \Delta^{\widetilde{G}} \varphi \rangle$. Specifically,*

$$\langle \varphi, \Delta^G \varphi \rangle = \sum_{u \sim_G v} |\varphi(u) - \varphi(v)|^2 w_{uv} = \sum_{u \sim_{\widetilde{G}} v} |\varphi(u) - \varphi(v)|^2 \widetilde{w}_{uv} = \langle \varphi, \Delta^{\widetilde{G}} \varphi \rangle.$$

*However, the addition of self-loops does affect the norm in the space of vertices:*

$$\langle \varphi, \varphi \rangle_{\ell_2(V,w)} = \sum_{v \in V} \varphi(v)^2 \deg_G(v) \leq \sum_{v \in V} \varphi(v)^2 \deg_{\widetilde{G}}(v) = \langle \varphi, \varphi \rangle_{\ell_2(V,\widetilde{w})}.$$

*Therefore, while the first quadratic form remains unchanged, the vertex norm in the extended space is generally increased due to the additional self-loops.*

## 3.2 A Cheeger-type Inequality for Constrained Clustering

Next we relate the constrained clustering problem to the generalized eigenvalue problem. Our key result is a Cheeger-type inequality proving that the value of $\Phi_H^G$ can be upper bounded with respect to $\lambda_2(\Delta_H^G)$ and $\lambda_2(\Delta^H)$, and the cut with the proven approximation guarantee can be found by a sweep-set algorithm. Our result is as follows:

**Theorem 1.** *Let $G = (V, E, w)$ and $H = (V, E', w')$ be graphs such that $\deg^G(v) = \deg^H(v)$ for all $v \in V$. Then, it holds that*

$$\Phi_H^G \leq 4 \sqrt{\frac{\lambda_2(\Delta_H^G)}{\lambda_2(\Delta^H)}}. \tag{4}$$

*where $\Delta^H$ denotes the normalized Laplacian of the graph $H$, $\Delta_H^G$ is the operator given by*

$$\Delta_H^G(x) := \frac{\langle x, \Delta^G x \rangle}{\langle x, \Delta^H x \rangle},$$

*and $\lambda_2$ is the smallest non-trivial eigenvalue of the corresponding operator defined in equation 2. Moreover, the cut achieving this approximation guarantee can be found with a sweep-set algorithm.*

Notice that we can assume that $G$ is a connected graph. If $H$ have $m$ connected component, we will work with $\lambda_{m-1}(\Delta^H)$, i.e. $x \perp \mathbf{1}_C$ for each connected component, ensuring $x \perp \ker(\Delta^H)$. Before presenting the proof, notice that we can assume without loss of generality that the graphs $G$ and $H$ have the same degree sequence due to the preprocessing step.

*Proof Sketch of Theorem 1.* This is a proof sketch; the complete proof is provided in the appendix.

**Step 1.** Let $x_1 \le x_2 \le \ldots \le x_n$ be the eigenvector associated with $\lambda_2(\Delta_H^G)$. It suffices to prove the existence of a non-empty, proper subset $\emptyset \subset S \subset V$ such that:

$$\frac{w_G(S, V \setminus S)}{w_H(S, V \setminus S)} = 4\sqrt{\frac{\langle x, \Delta^G x \rangle \cdot \langle x, x \rangle}{\langle x, \Delta^H x \rangle^2}}. \tag{5}$$

This suffices because the above inequality implies:

$$\Phi_H^G \le 4\sqrt{\lambda_2(\Delta_H^G)} \cdot \frac{\sqrt{\langle x, x \rangle}}{\sqrt{\langle x, \Delta^H x \rangle}} \le 4\sqrt{\lambda_2(\Delta_H^G)} \cdot \sqrt{\frac{1}{\lambda_2(\Delta^H)}}.$$

**Step 2.** We reduce the problem to proving equation 5 for a scaled version of $x$, defined as $z = cx$, where $z_n^2 + z_1^2 = 1$. The scaling ensures invariance of the expression:

$$\sqrt{\frac{\langle z, \Delta^G z \rangle}{\langle z, \Delta^H z \rangle}} \cdot \sqrt{\frac{\langle z, z \rangle}{\langle z, \Delta^H z \rangle}} = \sqrt{\frac{c^2 \langle x, \Delta^G x \rangle}{c^2 \langle x, \Delta^H x \rangle}} \cdot \sqrt{\frac{c^2 \langle x, x \rangle}{c^2 \langle x, \Delta^H x \rangle}} = \sqrt{\frac{\langle x, \Delta^G x \rangle}{\langle x, \Delta^H x \rangle}} \cdot \sqrt{\frac{\langle x, x \rangle}{\langle x, \Delta^H x \rangle}},$$

Thus, next equation is suffices to establish equation 5.:

$$\frac{w_G(S, V \setminus S)}{w_H(S, V \setminus S)} = 4\sqrt{\frac{\langle z, \Delta^G z \rangle \cdot \langle z, z \rangle}{\langle z, \Delta^H z \rangle^2}}, \tag{6}$$

**Step 3.** We now consider sweep sets $S_t$, defined as: $S_t = \{v \in V \mid z_v \le t\}$. To establish equation 6, it suffices to prove that there exists a threshold $t_0 \in [z_1, z_n]$ an defining $S = S_{t_0}$:

$$\frac{w_G(S_{t_0}, V \setminus S_{t_0})}{w_H(S_{t_0}, V \setminus S_{t_0})} \le 4\sqrt{\frac{\langle z, \Delta^G z \rangle}{\langle z, \Delta^H z \rangle}} \cdot \sqrt{\frac{\langle z, z \rangle}{\langle z, \Delta^H z \rangle}}. \tag{7}$$

**Step 4.** To establish equation 7, we will find a probability density function over $[z_1, z_n]$, and prove:

$$\frac{\mathbb{E}\left(w_G(S_t, V \setminus S_t)\right)}{\mathbb{E}\left(w_H(S_t, V \setminus S_t)\right)} \le 4\sqrt{\frac{\langle z, \Delta^G z \rangle \langle z, z \rangle}{\langle z, \Delta^H z \rangle^2}}. \tag{8}$$

Thus, equation 7 holds because the existence of a threshold $t_0$ is guaranteed by the next equation:

$$\frac{w_G(S_{t_0}, V \setminus S_{t_0})}{w_H(S_{t_0}, V \setminus S_{t_0})} \le \frac{\mathbb{E}\left(w_G(S_t, V \setminus S_t)\right)}{\mathbb{E}\left(w_H(S_t, V \setminus S_t)\right)} \implies \mathbb{P}\left\{\frac{w_G(S_{t_0}, V \setminus S_{t_0})}{w_H(S_{t_0}, V \setminus S_{t_0})} \le 4\sqrt{\frac{\langle z, \Delta^G z \rangle \langle z, z \rangle}{\langle z, \Delta^H z \rangle^2}}\right\} > 0.$$

**Step 5.** To establish equation 8, it suffices to find a probability distribution over $[z_1, z_n]$ such that:

$$\frac{\mathbb{E}\left(w_G(S_t, V \setminus S_t)\right)}{\mathbb{E}\left(w_H(S_t, V \setminus S_t)\right)} \le \frac{\sum_{u \sim_G v} |z_u - z_v|(|z_u| + |z_v|) w_{uv}}{\sum_{u \sim_H v} \frac{|z_u - z_v|^2}{2} w_{uv}}. \tag{9}$$

Using the Cauchy-Schwarz inequality, the definition of $\Delta$, and the property $\deg^G(v) = \deg^H(v)$, we derive equation 8 as follows:

$$\frac{\sum_{u \sim_G v} |z_u - z_v|(|z_u| + |z_v|) w_{uv}}{\sum_{u \sim_H v} \frac{|z_u - z_v|^2}{2} w_{uv}} \le \frac{\sqrt{\sum_{u \sim_G v} |z_u - z_v|^2 w_{uv} \sum_{u \sim_G v} (|z_u| + |z_v|)^2 w_{uv}}}{\frac{1}{2}\langle z, \Delta^H z \rangle} \le 4\sqrt{\frac{\langle z, \Delta^G z \rangle \langle z, z \rangle}{\langle z, \Delta^H z \rangle^2}}.$$

Thus, finding a distribution that satisfies equation 9 is sufficient to complete the proof.

**Step 6.** We define a distribution, and choose $t$ according to the probability density function $2|t|$. Specifically, the probability that a value between $[a, b]$ is chosen is

$$\mathbb{P}[t \in [z_v, z_u]] = \int_{z_v}^{z_u} 2|t|dt = \text{sgn}(z_u) \cdot z_u^2 - \text{sgn}(z_v) \cdot z_v^2.$$

Since $z_1^2 + z_n^2 = 1$, we have that $\mathbb{P}[t \in [z_1, z_n]] = 1$.

**Step 7.** For this distribution, and regardless of the sign of $z_u$ and $z_v$, we have:

$$\mathbb{E}\left[w_G(S_t, V \setminus S_t)\right] = \sum_{u \sim_G v} \mathbb{P}\left[z_u \leq t \text{ and } t < z_v\right] w_{uv} \leq \sum_{u \sim_G v} |z_u - z_v|(|z_u| + |z_v|)w_{uv},$$

$$\mathbb{E}\left[w_H(S_t, V \setminus S_t)\right] = \sum_{u \sim_H v} \mathbb{P}\left[z_u \leq t \text{ and } t < z_v\right] w_{uv} \geq \sum_{u \sim_H v} \frac{|z_u - z_v|^2}{2} w_{uv}.$$

This establishes equation 9 and conclude the proof. The complete details are in the appendix. $\square$

Based on the proof of Theorem 1, to find the cut with the guaranteed approximation, we only need to order the vertices based on the entries of the eigenvector for the generalised eigenvalue problem, and construct $n$ sweep sets. See Algorithm 1 for the formal description of our algorithm.

---

**Algorithm 1** The Constrained Clustering Algorithm

---

**Input:** Graph $G$ and graph $H$.
**Output:** A bi-partition of the vertex sets.
Compute the scaling factor $c_0$ defined in equation 3
Scale all edge weights in $G$ by multiplying them with $c_0$.
**for** each vertex $v \in V$ **do**
  **if** $\deg_H(v) > \deg_G(v)$ **then**
    Add a self-loop at $v$ with weight $\frac{1}{2}(\deg_H(v) - \deg_G(v))$.
  **end if**
**end for**
Compute the Laplacians $\Delta^G$ and $\Delta^H$ for the graphs $G$ and $H$.
Solve the generalized eigenvalue problem

$$\frac{\langle f, \Delta^G f \rangle}{\langle f, \Delta^H f \rangle} \quad \text{subject to} \quad f \perp \mathbf{1}, \tag{10}$$

where $f$ is the eigenvector that minimises the ratio.
Apply a sweep-set algorithm on the eigenvector $f$ to partition the vertices of $G$ into two clusters.
**Return:** a bi-partition of the vertex set

---

**Remark 2.** *Theorem 1 can be viewed as a generalization of the classical Cheeger inequality for graphs (Chung, 1997). Specifically, if we consider the graph $H$ as the complete graph with $w_{uv} = 1$ for all edges, then it is straightforward to show that*

$$\min_{\emptyset \subset S \subset V} \frac{w_G(S, V \setminus S)}{|S| \cdot |V \setminus S|} \leq 4\sqrt{\lambda_2(\Delta^G)},$$

*where $\lambda_2(\Delta^G)$ is the second smallest eigenvalue of the normalized graph Laplacian of $G$. Similarly, if we consider the graph $H = (V, E', w^H)$ as the complete graph with self-loops where $w_{uv}^H = \frac{\deg^G(u)\deg^G(v)}{\text{vol}(G)}$, then*

$$\min_{\emptyset \subseteq S \subseteq V} \frac{w_G(S, V \setminus S)}{\min(\text{vol}(S), \text{vol}(V \setminus S))} \leq \min_{\emptyset \subseteq S \subseteq V} \frac{\text{vol}(G)w_G(S, V \setminus S)}{\text{vol}(S)\text{vol}(V \setminus S)} \leq 4\sqrt{\lambda_2(\Delta^G)}.$$

### 3.3 PRACTICAL CONSIDERATIONS

The proof of Theorem 1 not only establishes the general cut bound but also provides a constructive method to find a subset $S \subseteq V$ that is close to minimizing the generalized cut problem. However, this approach can be computationally expensive, particularly because the Laplacian $H$ is not

invertible. To address this issue, we modify the graph $H$ by adding a "negative" self-loop at any vertex, effectively making the Laplacian invertible. This modification leverages the signed Laplacian, which adjusts the operator to ensure invertibility, and the introduction of a negative self-loop has little impact on the overall results, which will be demonstrated in Section 4 through experiments.

Formally, we prove that adding a negative self-loop to $H$ makes the signed Laplacian $\Delta_\alpha^{H'}$ invertible, ensuring $\lambda_1(\Delta_\alpha^{H'}) > 0$. Since $\lambda_1(\Delta_\alpha^{H'}) \le \lambda_2(\Delta^H)$, we can replace $\lambda_2(\Delta^H)$ with $\lambda_1(\Delta_\alpha^{H'})$ in Theorem 1, maintaining the theorem's validity.

**Lemma 2.** *Let $H = (V, E, w)$ be a weighted graph, and let $H' = (V, E', w')$ be another weighted graph such that $E' = E \cup \{(v_0, v_0)\}$, where $v_0$ is a vertex with a self-loop. Assume that $w'|_E = w$ and consider the signature $s = 1$ for all $E$ and $s_{(v_0, v_0)} = -1$. Then,*

$$0 < \lambda_1(\Delta_\alpha^{H'}) \le \lambda_2(\Delta^H),$$

*where $\Delta^H$ is the Laplacian of graph $H$.*

The proof of Lemma 2 shows that $\langle g, \Delta_\alpha^{H'} g \rangle \approx \langle g, \Delta^H g \rangle$ for a small weight in the self-loop, then we will solve equation 10 for the self-loop, because

$$\frac{\langle f, \Delta^G f \rangle}{\langle f, \Delta_\alpha^H f \rangle} \approx \frac{\langle f, \Delta^G f \rangle}{\langle f, \Delta_\alpha^{H'} f \rangle}.$$

The problem involves identifying the eigenfunction and eigenvalue of a linear operator using a Lagrangian-based framework. The Lagrangian $\mathcal{L}(\varphi, \lambda)$ is defined as

$$\mathcal{L}(\varphi, \lambda) = \langle \Delta^G \varphi, \varphi \rangle - \lambda(\langle \Delta_\alpha^{H'} \varphi, \varphi \rangle - 1),$$

where $\varphi$ is the function to be optimized, and $\lambda$ is the Lagrange multiplier. To find the minimizer $\varphi$, we set the gradient of $\mathcal{L}$ with respect to $\varphi$ to zero, i.e., $\nabla_\varphi \mathcal{L}(\varphi, \lambda) = 0$. Expanding this condition yields that $2\Delta^G \varphi - 2\lambda \Delta_\alpha^{H'} \varphi = 0$, which simplifies to $\Delta^G \varphi = \lambda \Delta_\alpha^{H'} \varphi$. This formulation leads to a generalized eigenvalue problem where $\varphi$ is the eigenfunction, and $\lambda$ is the eigenvalue. If $\Delta_\alpha^{H'}$ is invertible, the equation can be reformulated as

$$(\Delta_\alpha^{H'})^{-1} \Delta^G \varphi = \lambda \varphi,$$

illustrating the relationship between the linear operators and providing a solution to the eigenvalue problem via the Lagrange multiplier method. This eigenvalue equation is crucial for extracting the optimal partitions of the graph based on the constraints encoded within $\Delta_\alpha^{H'}$. We prove that solving the generalized eigenvalue problem for $\Delta^G$ and $\Delta_\alpha^{H'}$ produces all feasible solutions, as all eigenvalues are real and non-negative. This contrasts with the approach by Wang et al. (2014).

**Lemma 3.** *Let $\Delta_\alpha^{H'}$ be the signed Laplacian of the weighted graph $H'$, and $\Delta^G$ the normalized Laplacian of the weighted graph $G$. The operator $(\Delta_\alpha^{H'})^{-1} \Delta^G$ is a positive, self-adjoint operator, with all its eigenvalues are real and non-negative.*

We remark that solving equation 10 becomes significantly more efficient for $\Delta_\alpha^{H'}$ because it is symmetric positive definite and invertible. For dense matrices, this property allows for the use of the Cholesky decomposition with computational cost $O(n^3/3)$, reducing the problem to a standard eigenvalue problem (Saad, 2011). This is an improvement over the general case for the QZ algorithm (generalized Schur decomposition) where it is used with a complexity of $O(n^3)$. Moreover, the Cholesky decomposition enhances numerical stability, leading to fewer round-off errors. For large, sparse graphs and positive definite operators, iterative methods such as the Lanczos algorithm could be used. The complexity of these solvers are $O(nkm)$, where $k$ is the number of eigenvalues to find (smaller that $n$) and $m$ related with the iterations required for convergence and the condition number.

## 4 EXPERIMENTS

We conducted experiments to compare the spectral clustering method with the constrained clustering approach, using both synthetic and real-world datasets. The clustering accuracy was evaluated using the Adjusted Rand Index (**ARI**). All simulations were run on a PC equipped with an Intel® Core™ i7-10610U CPU running at 1.80 GHz and 32 GB of RAM, using MATLAB R2024a for computation. The three clustering algorithms compared in our experiments were:

- SPECTRAL CLUSTERING (SC): We computed the normalized Laplacian $\Delta^G$ and we used its second smallest eigenvector (Fiedler vector) for clustering the vertices.
- CONSTRAINED CLUSTERING (CC): We solved equation 10 and used the eigenvector corresponding to the smallest positive eigenvalue (excluding the trivial zero eigenvalue) for clustering.
- CONSTRAINED CLUSTERING WITH NEGATIVE SELF-LOOPS (CC++): Our algorithm consists in adding a negative self-loop and solve equation 10 for the signed Laplacian.

### 4.1 STOCHASTIC BLOCK MODEL

We considered a binary Stochastic Block Model (SBM) with $n = 1,000$ vertices divided into two equal-sized communities. Edges between vertices were generated based on intra-cluster probability $p$ and inter-cluster probability $q$. Specifically, we fixed $p = 0.2$ and varied $q$ from 0.12 to 0.2 in 30 equidistant steps. For each value of $q$, we generated two graphs generated from the SBM:

- $G$: a graph with intra-cluster edge probability $p$ and inter-cluster edge probability $q$.
- $H$: a graph generated with intra-cluster edge probability $q$ and inter-cluster edge probability $p$, effectively the complement of $G$ in terms of edge probabilities.

The vertex labels were kept consistent between $G$ and $H$. This variation allows us to observe how the clustering performance changes as the distinction between communities becomes less pronounced (since higher $q$ implies more inter-cluster edges). The experimental results, visualized in Figure 1a, illustrate the superior performance of CC++ compared to traditional SC, particularly as the inter-cluster edge probability $q$ increases.

At low values of $q$ (i.e., when the distinction between communities is clear), both methods perform well, achieving near-perfect ARI values. As seen in Figure 1a, both methods maintain ARI values close to 1.0 when $q \leq 0.14$. This is expected, as the strong intra-cluster edge probability $p = 0.2$ dominates the inter-cluster connections, making the community structure relatively easy to detect.

However, as $q$ increases, the performance of SC deteriorates rapidly. For instance, between $q = 0.16$ and $q = 0.18$, the ARI for SC drops sharply from approximately 0.7 to near 0.1. This decline occurs because higher values of $q$ increase the number of inter-cluster edges, blurring the distinction between communities. SC relies solely on the structure of $G$, struggles to correctly partition the vertices under these conditions.

In contrast, both of CC and CC++ are significantly more robust to increasing $q$. Even as $q$ approaches 0.17, the ARI remains above 0.5, significantly outperforming SC in this regime. This robustness stems from the ability of the Generalized Eigenvalue method to leverage the structural information of both $G$ and $H$, balancing the cuts between them, the method mitigates the negative impact of increased inter-cluster edges, thus maintaining better clustering accuracy even when the community structure is less pronounced.

This set of experimental results clearly demonstrates that the CC++ algorithm outperforms the traditional SC, particularly in challenging scenarios where the inter-cluster edge probability $q$ is high.

### 4.2 VARYING CLUSTER DISTANCE

Based on the Geometric Random Graph (RGG) (Avrachenkov et al., 2021; Dall & Christensen, 2002), we generated two clusters of vertices, each containing 500 points, randomly distributed within two-dimensional circular regions (disks) of radius 0.2. The separation between the clusters' centroids varied between -0.35 and 0.35, in 25 equidistant steps. This range allows us to simulate different levels of overlap between the clusters. Each configuration of cluster separation was repeated 10 times and we generated two graphs, $G$ and $H$, on the same set of vertices, but with different connectivity structures:

- $G$ : Vertices within the same cluster were connected with a large radius $r_{\text{intra}} = 0.1$, and vertices between clusters were connected with a smaller radius $r_{\text{inter}} = 0.05$.
- $H$ : The intra-cluster connection radius is reduced to $r_{\text{intra}} = 0.05$, while the inter-cluster radius is increased to $r_{\text{inter}} = 0.1$. Edges more likely connect vertices across the two clusters

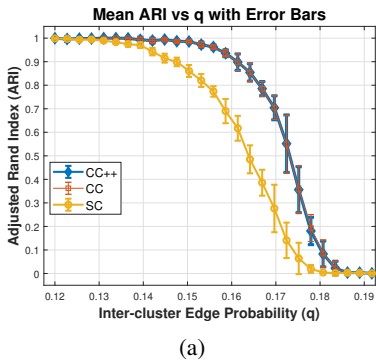
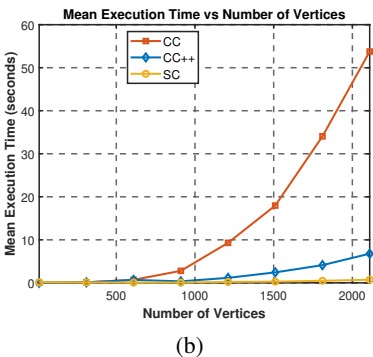

(a)  (b)

Figure 1: (a) Mean ARI vs Inter-cluster Edge Probability $q$ with error bars. CC (red) and CC++ (blue) consistently outperforms SC (yellow), especially as $q$ increases. (b) Mean execution time versus the number of vertices. The plot shows that CC++ (blue) performs similarly to SC (yellow) for smaller graphs but scales more efficiently than CC (red) as the number of vertices increases.

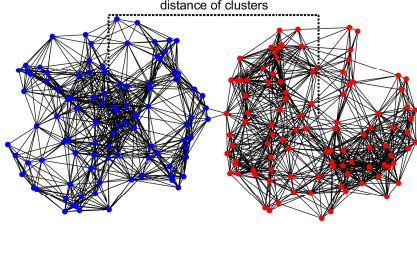
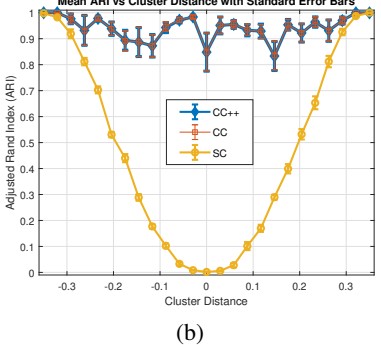

(a)  (b)

Figure 2: (a) Two clusters generated by a RGG with varying separation. (b) Comparison of ARI vs Cluster Distance for SC, CC, and CC++.

Figure 2a shows the RGG used in the experiments. The cluster distance is varied to simulate different levels of overlap between clusters. Figure 2b shows the ARI scores for both methods across different cluster distances. Each point represents the mean ARI, with error bars indicating standard error across 10 repetitions. When the cluster distance is large (above 0.3), both methods achieve near-perfect ARI values. The separation between clusters is clear, and both algorithms can detect the underlying structure accurately. As the clusters get closer, SC shows a significant drop in performance. For distances near zero, where clusters overlap, the ARI scores for SC drop to nearly zero, indicating its struggle to differentiate overlapping clusters. In contrast, the Generalized Eigenvalue method is more resilient to cluster overlap, maintaining significantly higher ARI values even when the clusters become indistinguishable by conventional means. This robustness is due to the method's ability to leverage information from both graphs $G$ and $H$, capturing both intra- and inter-cluster relationships.

## 4.3 EXPERIMENTS WITH TEMPERATURE DATA

We evaluated SC and CC++ on real-world temperature data from ground stations in Brittany, January 2014 (Girault, 2015). The experiments considered three data types: temperature, maximal temperature, and minimal temperature. Temperature values con be seen as graph signals (Ortega et al., 2018), with vertices representing readings. The aim is to cluster stations by proximity and similar temperature patterns, combining spatial and temperature data. This approach can be applied

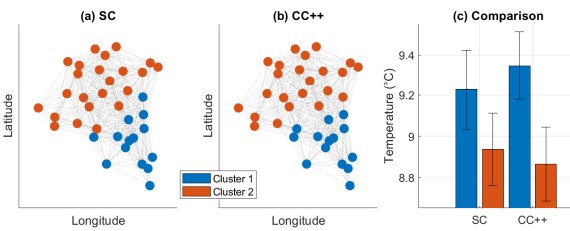

Figure 3: (a) Clustering based on SC, grouping stations by geographical proximity. (b) Clustering using CC++, which considers both location and temperature similarity. (c) Mean and SE of temperature, illustrating no overlap for CC++ compared to SC.

Table 1: Percentage of successful separation of regions using temperature and location data.

| Data Type | SC (%) | CC (%) | CC++ (%) |
|---|---|---|---|
| Temperature | 63.30% | 79.16% | 79.16% |
| Maximal Temperature | 62.90% | 80.91% | 81.04% |
| Minimal Temperature | 62.63% | 79.16% | 77.95% |

to identify micro climates (Cao et al., 2021) or to segment regions for agricultural (Yao et al., 2022), where both location and temperature are important factors.

We construct a graph from the input data as follows: every station is represented as a vertex, and the edge weight is defined by spatial similarity via a Gaussian kernel: $W_{ij} = \exp\left(-\frac{d(i,j)^2}{2\sigma_1^2}\right)$ if $d(i,j) \leq \sigma_2$, and 0 otherwise, where $d(i,j)$ is the Euclidean distance. Parameters were set to $\sigma_1^2 = 5\times10^8$ and $\sigma_2 = 10^5$ (Girault, 2015). SC primarily grouped stations by proximity (Figure 3a).

To construct the constraint graph $H$, we use the temperature values and an inverse Gaussian kernel, assigning edge weights close to 1 for large temperature differences and 0 for similar temperatures, analogous to how $G$ is built using spatial proximity. This allows the clustering algorithm to incorporate both geographical and temperature constraints for a more nuanced partition. The output of CC++ is shown in Figure 3b, where only two cities differ from the clustering based on spatial proximity alone (SC). Finally, we compute the mean and standard error (SE) for each cluster and method. For this specific hour, SC shows overlap between clusters' temperature values, whereas CC++ achieves no overlap, indicating better clustering of stations with similar temperatures Figure 3c.

We repeated this process for all available measurements, one per hour, across 24 hours over 31 days, totaling 744 observations. To evaluate clustering accuracy, we calculated the mean and SE for clustering using both methods. A separation was considered correct if the clusters' values did not overlap, as determined by their SE. Table 1 summarizes the results, showing the percentage of correct separations for each temperature data type when comparing SC and CC++.

As shown in Table 1, CC++ consistently outperformed CC across all temperature data types. This can be attributed to the method's ability to leverage both the spatial structure of the stations and the actual temperature data. In contrast, SC, which relies solely on the graph structure, struggled to accurately separate stations into distinct clusters when the temperature differences were subtle.

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

## A  OMITTED DETAILS FROM SECTION 3

This section presents the details omitted from Section 3.

*Proof of Theorem 1.* We aim to prove the inequality:

$$\Phi_H^G \leq 4\sqrt{\frac{\lambda_2(\Delta_H^G)}{\lambda_2(\Delta^H)}}. \tag{11}$$

**Step 1.** Let $x : V \to \mathbb{R}$ be the eigenfunction associated with $\lambda_2(\Delta_H^G)$. Without loss of generality, we renumber the vertices such that the coordinates of $x$ satisfy $x_1 \leq x_2 \leq \ldots \leq x_n$. It suffices to prove the existence of a non-empty, proper subset $\emptyset \subset S \subset V$ such that:

$$\frac{w_G(S, V \setminus S)}{w_H(S, V \setminus S)} = 4\sqrt{\frac{\langle x, \Delta^G x\rangle \cdot \langle x, x\rangle}{\langle x, \Delta^H x\rangle^2}}. \tag{12}$$

This is sufficient because:

$$\Phi_H^G \leq \frac{w_G(S, V \setminus S)}{w_H(S, V \setminus S)} \qquad \text{(Definition of } \Phi_H^G \text{ as the minimum ratio cut)}$$

$$= 4\sqrt{\frac{\langle x, \Delta^G x\rangle}{\langle x, \Delta^H x\rangle}} \cdot \sqrt{\frac{\langle x, x\rangle}{\langle x, \Delta^H x\rangle}} \qquad \text{(By equation 12)}$$

$$= 4\sqrt{\lambda_2(\Delta_H^G)} \cdot \frac{\sqrt{\langle x, x\rangle}}{\sqrt{\langle x, \Delta^H x\rangle}} \qquad \text{(Using } x \text{ is the eigenvector of } \lambda_2(\Delta_H^G))$$

$$\leq 4\sqrt{\lambda_2(\Delta_H^G)} \cdot \sqrt{\frac{1}{\lambda_2(\Delta^H)}} \qquad \text{(Since } \frac{\langle x, \Delta^H x\rangle}{\langle x, x\rangle} \geq \lambda_2(\Delta^H)).$$

This inequality proves that establishing equation 12 suffices to derive equation 11.

**Step 2.** We now show that proving equation 12 for $x$ is equivalent to proving the existence of $S$ such that :

$$\frac{w_G(S, V \setminus S)}{w_H(S, V \setminus S)} = 4\sqrt{\frac{\langle z, \Delta^G z\rangle \langle z, z\rangle}{\langle z, \Delta^H z\rangle^2}}, \tag{13}$$

where $z = cx$ is a scaled version of $x$, defined such that $z_n^2 + z_1^2 = 1$.

Because $x \perp \mathbf{1}$, we have $x_1 < 0 < x_n$, and thus scaling $x$ does not affect the ratio. Specifically:

$$\sqrt{\frac{\langle z, \Delta^G z\rangle}{\langle z, \Delta^H z\rangle} \cdot \frac{\langle z, z\rangle}{\langle z, \Delta^H z\rangle}} = \sqrt{\frac{c^2\langle x, \Delta^G x\rangle}{c^2\langle x, \Delta^H x\rangle} \cdot \frac{c^2\langle x, x\rangle}{c^2\langle x, \Delta^H x\rangle}} = \sqrt{\frac{\langle x, \Delta^G x\rangle}{\langle x, \Delta^H x\rangle} \cdot \frac{\langle x, x\rangle}{\langle x, \Delta^H x\rangle}}.$$

Thus, proving equation 13 suffices to establish equation 12.

**Step 3.** Let $t \in \mathbb{R}$, and define the set:

$$S_t = \{v \in V \mid z_v \leq t\},$$

commonly referred to as a sweep set. To establish equation 13, it suffices to prove that there exists a threshold $t_o$ such that:

$$\frac{w_G(S_{t_o}, V \setminus S_{t_o})}{w_H(S_{t_o}, V \setminus S_{t_o})} \leq 4\sqrt{\frac{\langle z, \Delta^G z\rangle}{\langle z, \Delta^H z\rangle}} \cdot \sqrt{\frac{\langle z, z\rangle}{\langle z, \Delta^H z\rangle}}. \tag{14}$$

Establishing $S = S_{t_0}$, equation 14 directly implies equation 13.

**Step 4.** To establish equation 14, it suffices to show that there exists a distribution $f(t)$ over $[z_1, z_n]$ such that the expectation satisfies:

$$\frac{\mathbb{E}\left(w_G(S_t, V \setminus S_t)\right)}{\mathbb{E}\left(w_H(S_t, V \setminus S_t)\right)} \leq 4\sqrt{\frac{\langle z, \Delta^G z \rangle \langle z, z \rangle}{\langle z, \Delta^H z \rangle^2}}. \tag{15}$$

Here, $S_t$ is treated as a random variable parameterized by $t \in [z_1, z_n]$, where $t$ is drawn from a probability density function $f(t)$. $S_t$ depends on the distribution of $t$, which influences the likelihood of including a vertex $v$ in $S_t$ based on its value $z_v$.

This suffices because, if we define

$$\varphi = \frac{\mathbb{E}\left(w_G(S_t, V \setminus S_t)\right)}{\mathbb{E}\left(w_H(S_t, V \setminus S_t)\right)} \in \mathbb{R},$$

then, by linearity of expectation:

$$\mathbb{E}\left[w_G(S_t, V \setminus S_t) - \varphi\, w_H(S_t, V \setminus S_t)\right] = \mathbb{E}\left[w_G(S_t, V \setminus S_t)\right] - \varphi\, \mathbb{E}\left[w_H(S_t, V \setminus S_t)\right] = 0.$$

This implies that for some $t_0$:

$$\mathbb{P}\left[w_G(S_{t_o}, V \setminus S_{t_o}) - \varphi\, w_H(S_{t_o}, V \setminus S_{t_o}) \leq 0\right] > 0, \tag{16}$$

because if for all $t$:

$$\mathbb{P}\left[w_G(S_t, V \setminus S_t) - \varphi\, w_H(S_t, V \setminus S_t) \leq 0\right] = 0,$$

it would follow that $w_G(S_t, V \setminus S_t) - \varphi\, w_H(S_t, V \setminus S_t) > 0$ for all $t$, contradicting the fact that:

$$\mathbb{E}\left[w_G(S_t, V \setminus S_t) - \varphi\, w_H(S_t, V \setminus S_t)\right] = 0.$$

Thus, equation 16 holds, which implies that for some $t_0$:

$$w_G(S_{t_0}, V \setminus S_{t_0}) - \varphi\, w_H(S_{t_0}, V \setminus S_{t_0}) \leq 0,$$

and equivalently:

$$\frac{w_G(S_{t_0}, V \setminus S_{t_0})}{w_H(S_{t_0}, V \setminus S_{t_0})} \leq \frac{\mathbb{E}\left(w_G(S_t, V \setminus S_t)\right)}{\mathbb{E}\left(w_H(S_t, V \setminus S_t)\right)}.$$

Combining this with equation 16, we obtain:

$$\mathbb{P}\left\{\frac{w_G(S_{t_0}, V \setminus S_{t_0})}{w_H(S_{t_0}, V \setminus S_{t_0})} \leq \frac{\mathbb{E}\left(w_G(S_t, V \setminus S_t)\right)}{\mathbb{E}\left(w_H(S_t, V \setminus S_t)\right)}\right\} > 0.$$

From Eq. equation 15, it follows that:

$$\mathbb{P}\left\{\frac{w_G(S_{t_0}, V \setminus S_{t_0})}{w_H(S_{t_0}, V \setminus S_{t_0})} \leq 4\sqrt{\frac{\langle z, \Delta^G z \rangle \langle z, z \rangle}{\langle z, \Delta^H z \rangle^2}}\right\} > 0.$$

Establishing equation 15 thus guarantees the existence of $t$ satisfying equation 14.

**Step 5.** To establish equation 15, it suffices to find a probability distribution over $[z_1, z_n]$ such that the following inequality holds:

$$\frac{\mathbb{E}\left(w_G(S_t, V \setminus S_t)\right)}{\mathbb{E}\left(w_H(S_t, V \setminus S_t)\right)} \leq \frac{\sum_{u \sim_G v} |z_u - z_v|(|z_u| + |z_v|)w_{uv}}{\sum_{u \sim_H v} \frac{|z_u - z_v|^2}{2} w_{uv}}. \tag{17}$$

We now show that equation 17 implies equation 15 as follows:

$$\frac{\mathbb{E}\left(w_G(S_t, V \setminus S_t)\right)}{\mathbb{E}\left(w_H(S_t, V \setminus S_t)\right)} \leq \frac{\sum_{u \sim_G v} |z_u - z_v|(|z_u| + |z_v|)w_{uv}}{\sum_{u \sim_H v} \frac{|z_u - z_v|^2}{2} w_{uv}} \qquad \text{(By equation 17)}$$

$$\leq \frac{\sqrt{\sum_{u \sim_G v} |z_u - z_v|^2 w_{uv}} \cdot \sqrt{\sum_{u \sim_G v} (|z_u| + |z_v|)^2 w_{uv}}}{\sum_{u \sim_H v} \frac{|z_u - z_v|^2}{2} w_{uv}} \qquad \text{(Cauchy-Schwarz inequality)}$$

$$= \frac{\sqrt{\langle z, \Delta^G z \rangle} \cdot \sqrt{\sum_{u \sim_G v} (|z_u| + |z_v|)^2 w_{uv}}}{\sum_{u \sim_H v} \frac{|z_u - z_v|^2}{2} w_{uv}} \qquad \text{(Definition of } \Delta^G)$$

$$\leq \frac{\sqrt{\langle z, \Delta^G z \rangle} \cdot \sqrt{2 \sum_{u \sim_G v} (z_u^2 + z_v^2) w_{uv}}}{\sum_{u \sim_H v} \frac{|z_u - z_v|^2}{2} w_{uv}} \qquad \text{(Since } (a + b)^2 \leq 2(a^2 + b^2))$$

$$= \frac{\sqrt{\langle z, \Delta^G z \rangle} \cdot \sqrt{4 \langle z, z \rangle_{\ell_2(G)}}}{\sum_{u \sim_H v} \frac{|z_u - z_v|^2}{2} w_{uv}} \qquad \text{(Definition of the inner product in } G)$$

$$= \frac{\sqrt{\langle z, \Delta^G z \rangle} \cdot 2\sqrt{\langle z, z \rangle_{\ell_2(G)}}}{\frac{1}{2} \langle z, \Delta^H z \rangle} \qquad \text{(Definition of } \Delta^H)$$

$$\leq 4\sqrt{\frac{\langle z, \Delta^G z \rangle \langle z, z \rangle_{\ell_2(H)}}{\langle z, \Delta^H z \rangle^2}} \qquad \text{(Since } \deg^G(v) = \deg^H(v)).$$

Thus, equation 17 implies equation 15.

**Step 6.** Consider the non-negative function $2|t|$ defined on the interval $[z_1, z_n]$. This function serves as a probability density function (PDF) over $[z_1, z_n]$ because $z_1^2 + z_n^2 = 1$, $z_1$ is negative, and $z_n$ is positive. Specifically:

$$\int_{z_1}^{z_n} 2|t| \, dt = \text{sgn}(z_n) \cdot z_n^2 - \text{sgn}(z_1) \cdot z_1^2 = 1.$$

The normalization in step 2 ensures that the integral of $2|t|$ over $[z_1, z_n]$ equals 1, validating it as a PDF. Hence, the probability that a value between $[z_u, z_v]$ is given by

$$\mathbb{P}[t \in [z_v, z_u]] = \int_{z_v}^{z_u} 2|t| dt = \text{sgn}(z_u) \cdot z_u^2 - \text{sgn}(z_v) \cdot z_v^2.$$

The expectation $\mathbb{E}\left[w_G(S_t, V \setminus S_t)\right]$ represents the expected weight of the cut $w_G(S_t, V \setminus S_t)$, which depends on the random threshold $t$ sampled from the previously defined probability density function. By the linearity of expectation:

$$\mathbb{E}\left[w_G(S_t, V \setminus S_t)\right] = \sum_{u \sim_G v} \mathbb{E}\left[\mathbf{1}_{u \in S_t \text{ and } v \notin S_t}\right] w_{uv},$$

where $\mathbf{1}_{u \in S_t \text{ and } v \notin S_t}$ is the indicator function that equals 1 when $u \in S_t$ and $v \notin S_t$, and 0 otherwise.

Using the definition of probability, the expectation simplifies to:

$$\mathbb{E}\left[w_G(S_t, V \setminus S_t)\right] = \sum_{u \sim_G v} \mathbb{P}\left[z_u \leq t \text{ and } t < z_v\right] w_{uv}.$$

If we can establish the following bounds:

$$\frac{|z_u - z_v|^2}{2} \leq \mathbb{P}\left[z_u \leq t \text{ and } t < z_v\right] \leq (|z_u| + |z_v|)|z_u - z_v|, \qquad (18)$$

then we can bound the expectation of the weight of the cut as follows:

$$\mathbb{E}\left[w_G(S_t, V \setminus S_t)\right] \leq \sum_{u \sim_G v} (|z_u| + |z_v|)|z_u - z_v|w_{uv}.$$

Similarly, for $H$, we have:

$$\mathbb{E}\left[w_H(S_t, V \setminus S_t)\right] = \sum_{u \sim_H v} \mathbb{P}\left[z_u \leq t \text{ and } t < z_v\right]w_{uv} \geq \frac{1}{2}\sum_{u \sim_H v} |z_u - z_v|^2 w_{uv}.$$

These last two inequalities establish equation 17. Therefore, to conclude the proof, it remains to prove equation 18.

**Step 7.** To prove equation 18, recall that:

$$\mathbb{P}\left[z_u \leq t \text{ and } t < z_v\right] = \int_{z_v}^{z_u} 2|t| \, dt = \begin{cases} |z_u^2 - z_v^2| & \text{if } \operatorname{sgn}(z_u) = \operatorname{sgn}(z_v), \\ z_u^2 + z_v^2 & \text{if } \operatorname{sgn}(z_u) \neq \operatorname{sgn}(z_v). \end{cases}$$

We first establish the upper bound in equation 18:

$$\mathbb{P}\left[z_u \leq t \text{ and } t < z_v\right] \leq (|z_u| + |z_v|)|z_u - z_v|. \tag{19}$$

*Case 1: $sgn(z_u) = sgn(z_v)$.* In this case:

$$|z_u^2 - z_v^2| = |(z_u + z_v)(z_u - z_v)| = |z_u + z_v||z_u - z_v|.$$

Since $|z_u + z_v| \leq |z_u| + |z_v|$, we have:

$$|z_u^2 - z_v^2| \leq (|z_u| + |z_v|)|z_u - z_v|.$$

*Case 2: $sgn(z_u) \neq sgn(z_v)$.* In this case:

$$z_u^2 + z_v^2 \leq (z_u - z_v)^2 = |z_u - z_v|^2,$$

and thus:

$$z_u^2 + z_v^2 \leq (|z_u| + |z_v|)|z_u - z_v|.$$

Combining both cases establishes the upper bound in equation 19.

Now, we establish the lower bound in equation 18:

$$\frac{|z_u - z_v|^2}{2} \leq \mathbb{P}\left[z_u \leq t \text{ and } t < z_v\right]. \tag{20}$$

*Case 1: $sgn(z_u) = sgn(z_v)$.* In this case:

$$\left|z_u^2 - z_v^2\right| = |z_u - z_v| \, |z_u + z_v|.$$

Since $|z_u + z_v| \geq |z_u - z_v|$, we have:

$$\left|z_u^2 - z_v^2\right| \geq |z_u - z_v|^2 \geq \frac{(z_u - z_v)^2}{2}.$$

*Case 2: $sgn(z_u) \neq sgn(z_v)$.* Using:

$$0 \leq (z_u + z_v)^2 = 2(z_u^2 + z_v^2) - (z_u - z_v)^2,$$

it follows that:

$$\frac{(z_u - z_v)^2}{2} \leq z_u^2 + z_v^2.$$

Combining both cases establishes the lower bound in equation 20.

Finally, combining equation 19 and equation 20 proves equation 18, completing the proof. $\qquad \square$

*Proof of Lemma 2.* Let $f$ be the eigenfunction corresponding to $\lambda_2(\Delta^H)$. Consider the function $g : V \to \mathbb{R}$, defined as: $g_v = f_v - f_{v_0}$, for all $v \in V$, hence $g_{v_0} = 0$. By computing the Rayleigh quotient for $g$ with respect to the Laplacian of $H$:

$$\langle g, \Delta^H g \rangle_H = \sum_{u \sim_H v} |g_u - g_v|^2 w_{uv} = \sum_{u \sim_H v} |f_u - f_v|^2 w_{uv} = \lambda_2(\Delta^H) \langle f, f \rangle_H.$$

The Rayleigh quotient for $g$ with respect to $\Delta^{H'}$ is given by

$$\langle g, \Delta^{H'} g \rangle_{H'} = \sum_{u \sim_{H'} v} |g_u - g_v|^2 w'_{uv} + 2|g_{v_0}|^2 w'_{v_0 v_0} = \sum_{u \sim_H v} |f_u - f_v|^2 w_{uv} = \langle g, \Delta^H g \rangle_H.$$

The norm of $g$ with respect to $H'$ satisfies that

$$\langle g, g \rangle_{H'} = \sum_{v \in V} |g_v|^2 \deg^{H'}(v) = \sum_{v \in V} |g_v|^2 \deg^H(v) + g_{(v_0)}^2 (\deg_H(v_0) + 2) = \langle g, g \rangle_H.$$

Thus, we have, since $f$ is an eigenfunction $f \perp \mathbf{1}$, hence $0 = \langle f, \mathbf{1} \rangle_H$:

$$\langle g, g \rangle_H = \sum_{v \in V} |f_v - f_{v_0}|^2 \deg^H(v) = \langle f, f \rangle_H - 2 f_{v_0} \langle f, \mathbf{1} \rangle_H + f_{v_0}^2 \langle \mathbf{1}, \mathbf{1} \rangle_H \geq \langle f, f \rangle_H.$$

Using this, we apply the Rayleigh quotient to bound $\lambda_1(\Delta_\alpha^{H'})$ as

$$\lambda_1(\Delta_\alpha^{H'}) \leq \frac{\langle g, \Delta^{H'} g \rangle_{H'}}{\langle g, g \rangle_{H'}} \leq \frac{\langle g, \Delta^H g \rangle_H}{\langle g, g \rangle_H} \leq \frac{\lambda_2(\Delta^H) \langle f, f \rangle_H}{\langle f, f \rangle_H} = \lambda_2(\Delta^H).$$

Finally, let $f$ be a non-zero function. Consider the following expression:

$$\langle f, \Delta_\alpha^{H'} f \rangle = \sum_{u \sim v} |f_u - f_v|^2 w_{uv} + 2 f_{v_0}^2 w_{v_0 v_0}.$$

We have $0 \leq \langle f, \Delta_\alpha^{H'} f \rangle$, and equality holds if and only if: $\langle f, \Delta_\alpha^{H'} f \rangle = 0 \iff f_u - f_v = 0 \quad \forall u, v$ and $f_{v_0} = 0$. This implies that $f = 0$ for all $v \in V$, which contradicts the assumption that $f$ is a non-zero function. Therefore, $\lambda_1(\Delta_\alpha^{H'}) > 0$. $\square$

*Proof of Lemma 3.* Since $\Delta_\alpha^{H'}$ is positive-definite and self-adjoint, its inverse $(\Delta_\alpha^{H'})^{-1}$ and its square root $(\Delta_\alpha^{H'})^{1/2}$ exist and are self-adjoint operators. Define the operator $C$ as:

$$C = (\Delta_\alpha^{H'})^{-1/2} \Delta^G (\Delta_\alpha^{H'})^{-1/2}.$$

*Self-adjointness of $C$:* The adjoint of $C$ is:

$$C^* = \left( (\Delta_\alpha^{H'})^{-1/2} \Delta^G (\Delta_\alpha^{H'})^{-1/2} \right)^* = (\Delta_\alpha^{H'})^{-1/2} (\Delta^G)^* (\Delta_\alpha^{H'})^{-1/2}$$
$$= (\Delta_\alpha^{H'})^{-1/2} \Delta^G (\Delta_\alpha^{H'})^{-1/2} = C,$$

since $\Delta_\alpha^{H'}$ and $\Delta^G$ are self-adjoint operators.

*Positive Semi-definiteness of $C$:* For any function $\varphi \in \ell_2(V, w)$, consider:

$$\langle C\varphi, \varphi \rangle = \left\langle (\Delta_\alpha^{H'})^{-1/2} \Delta^G (\Delta_\alpha^{H'})^{-1/2} \varphi, \varphi \right\rangle.$$

Let $\psi = (\Delta_\alpha^{H'})^{-1/2} \varphi$. Then, $\langle C\varphi, \varphi \rangle = \langle \Delta^G \psi, \psi \rangle$. Since $\Delta^G$ is positive semi-definite, we have $\langle \Delta^G \psi, \psi \rangle \geq 0$. Therefore, $\langle C\varphi, \varphi \rangle \geq 0$, which means $C$ is positive semi-definite.

Because $C$ is self-adjoint and positive semi-definite, it is diagonalizable with real, non-negative eigenvalues. Thus, there exists an orthogonal matrix $W$ and a diagonal matrix $\Lambda$ with non-negative entries such that:

$$C = W \Lambda W^*.$$

We can express $(\Delta_\alpha^{H'})^{-1}\Delta^G$ as:

$$(\Delta_\alpha^{H'})^{-1}\Delta^G = (\Delta_\alpha^{H'})^{-1/2}\left((\Delta_\alpha^{H'})^{-1/2}\Delta^G(\Delta_\alpha^{H'})^{-1/2}\right)(\Delta_\alpha^{H'})^{1/2}$$

$$= (\Delta_\alpha^{H'})^{-1/2}C(\Delta_\alpha^{H'})^{1/2} = (\Delta_\alpha^{H'})^{-1/2}W\Lambda W^\top(\Delta_\alpha^{H'})^{1/2}.$$

Let $V = (\Delta_\alpha^{H'})^{-1/2}W$. Then,

$$(\Delta_\alpha^{H'})^{-1}\Delta^G = V\Lambda V^*.$$

Since $V$ is invertible (as the product of invertible matrices), $(\Delta_\alpha^{H'})^{-1}\Delta^G$ is diagonalizable with real, non-negative eigenvalues.

This completes the proof. $\qquad\square$

## B  ADDITIONAL EXPERIMENTS FROM SECTION 4

In this section, we present additional experiments to evaluate the performance of our algorithm. To provide a comprehensive analysis, we include comparisons with additional methods. This section will be further updated with more comparative methods as we expand our experiments. The method included in this version is:

- FLEXIBLE CONSTRAINED SPECTRAL CLUSTERING (FC): This method is presented in Wang & Davidson (2010).

As in Subsection 4.1, we evaluate the performance of clustering methods using synthetic graphs generated by the stochastic block model (SBM). We analyze the algorithms for graphs of varying sizes, focusing particularly on smaller graphs, where subtle variations are more prominent. For larger graphs, the performance trends tend to stabilize and exhibit fewer differences. In these experiments, graphs were generated with the number of nodes $n$ varying across four sizes: $n = 250, 500, 750, 1000$.

The results are summarized in Figure 4, which illustrates the performance of the four clustering methods—Spectral Clustering (SC), Constrained Clustering (CC), Constrained Clustering with Self-loops (CC++), and Flexible Clustering (FC)—for varying inter-cluster edge probabilities $q$ and different graph sizes.

Based on the results presented in Figure 4, we highlight the following key observations and advantages of our method compared to the baseline approaches:

*Improved Performance on Smaller Graphs:* Our method demonstrates superior performance on smaller graphs ($n = 250, 500$) in terms of the mean Adjusted Rand Index (ARI), as shown in panels (a) and (b). As the graph size increases ($n = 750, 1000$), the performance of our approach becomes comparable to that of the other methods, indicating that our algorithm is robust across different graph sizes.

*Parameter-Free Advantage:* The Flexible Clustering (FC) method presented in Wang & Davidson (2010) requires the user to define an additional parameter ($\beta$) that directly influences the solution of the generalized eigenvalue problem. This parameter must be carefully chosen to ensure at least one feasible solution exists, as incorrect parameter selection can result in negative eigenvalues and infeasible outcomes. In contrast, our method avoids this issue entirely. As shown in Lemma 3, the introduction of self-loops ensures that the operator $(\Delta_\alpha^{H'})^{-1}\Delta^G$ is positive and self-adjoint, with all eigenvalues guaranteed to be real and non-negative.

*Cheeger-type inequality:* Unlike the FC method, which does not provide a theoretical guarantee linking the eigenfunctions used for clustering to the optimization objective, our approach establishes a Cheeger-type inequality.

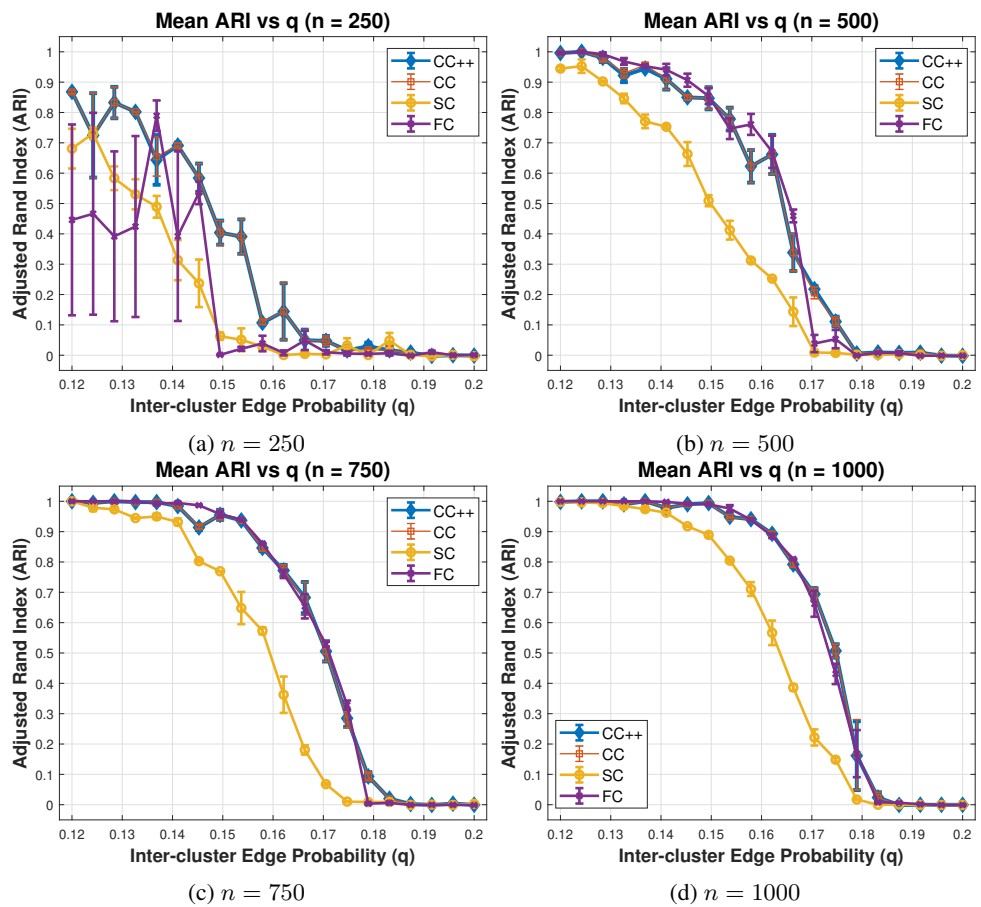

Figure 4: Mean Adjusted Rand Index (ARI) as a function of inter-cluster edge probability $q$ for four clustering methods. Each panel represents a different graph size: (a) $n = 250$, (b) $n = 500$, (c) $n = 750$, and (d) $n = 1000$.

