# OpenReview forum: "Constrained Graph Clustering with Signed Laplacians"
_ICLR.cc/2025/Conference — Submitted to ICLR 2025_

### Official Review · Reviewer_hnYU · 2024-10-23

**Soundness:** 1
**Presentation:** 3
**Contribution:** 2
**Rating:** 3
**Confidence:** 4

**Summary:**

This paper addresses the constrained clustering problem, which involves partitioning data into k disjoint clusters, subject to both "must-link" constraints (represented by a graph G) and "cannot-link" constraints (represented by a graph H). The objective is to minimize the number of G-edges across clusters while maximizing the number of H-edges within clusters. Several variants of this problem exist, depending on the specific settings, such as noisy versus hard constraints. The authors focus on a restricted version where k=2 and the constraints are hard. They propose a relaxation of the ratio-cut formulation, which leads to a generalized eigenvalue problem. In addition, they derive a Cheeger-type inequality for this ratio cut.

**Strengths:**

The numerical results provided in the paper are promising and suggest that the proposed approach performs well in practice. The idea of adding negative self-loops appears to significantly decrease the running time.

**Weaknesses:**

* Given the extensive literature on clustering with signed graphs and constrained clustering, the related work section is underdeveloped. The authors should provide a more detailed explanation of how their approach differs from existing methods in constrained spectral clustering, particularly in comparison to the algorithm presented in [1].

* The proof of Theorem 1 contains significant flaws, which must be addressed.

* The numerical evaluation lacks a strong baseline comparison.  Indeed, the authors only compare against an algorithm that uses only G (and not H). Including a comparison with other constrained spectral clustering methods (for example from references [1,2,3]), would convince the algorithm of the paper (which I believe compared to [1], the main difference is the addition of negative self-loops).

* Using spectral clustering as the baseline on geometric graphs is not entirely appropriate. The model presented in Section 4.2 is an instance of the Geometric Block Model (GBM), which has gained popularity in recent graph clustering literature. However, as shown in [4], when the distance between cluster centers is 0, standard spectral clustering using the second eigenvector fails to cluster geometric graphs. But, alternative algorithms, such as triangle counting or those using different eigenvectors, can succeed in such settings. Therefore, the accuracy drop observed in Figure 2b may be attributed to the choice of spectral clustering, while another clustering algorithm might have succeeded by using only G. Specifically, even when the distance between centers is 0, the communities may still be recoverable from GG alone, given that $r_{inter} \ne r_{intra}$.


References:

[1] Mihai Cucuringu, Ioannis Koutis, Sanjay Chawla, Gary Miller, and Richard Peng. Simple and scalable constrained clustering: a generalized spectral method. In Artificial Intelligence and Statistics, pp. 445–454, 2016.

[2] Jianyuan Li, et al. "Scalable constrained spectral clustering." IEEE Transactions on Knowledge and Data Engineering 27.2 (2014): 589-593.

[3] Xiang Wang, Buyue Qian, and Ian Davidson. "On constrained spectral clustering and its applications." Data Mining and Knowledge Discovery 28 (2014): 1-30.

[4] Konstantin Avrachenkov, Andrei Bobu, and Maximilien Dreveton. "Higher-order spectral clustering for geometric graphs." Journal of Fourier Analysis and Applications 27.2 (2021): 22.

**Questions:**

Specific Issues:

*    Line 226: The phrase "we prove this implies that..." lacks clarity, as I believe the proof of this implication is not given anywhere.
*    Line 241: The probability mentioned here is unclear. The authors should specify what this probability refers to.
*    Line 243: The claim that "the events are independent" requires clarification—what specific events are being referenced?
*    Line 248: The term "we define a distribution" is vague. The authors should clarify what distribution is being defined and its relevance to the analysis.
*    Line 272: The statement "Combining equation 4 with equation 5 proves the result" is incorrect. Even if the equations were correctly proven, this would yield a ratio of expectations rather than the ratio cut. Moreover, the expectation of a ratio is not necessarily equivalent to the ratio of expectations.
*    Line 322: I am unsure where does the inequality $\lambda_1(\Delta^{H'}_{\alpha}) \le \lambda_2(\Delta^H)$ comes from?


Additional Comments:

*    The proof of Theorem 1 is repeated in the appendix.
*    Line 57: The statement "the running time of our algorithm is close to traditional spectral clustering methods" is misleading. Figure 1b indicates a substantial deviation in running time between CC++ and SC when n becomes large. Additionally, the figure should display results for larger values of n to assess the evolution of the gap between the running times. 2000 vertices still correspond to small graphs.
*    Line 277: The phrase "to find the cut with guaranteed approximation" is ambiguous: what approximation guarantee is being referenced?
*   Line 278: "n sweep sets": what does a sweep set mean?
*    Line 290: It would be helpful for the authors to provide more details on how the generalized eigenvalue problem arises, even if it is not a novel contribution (I believe this has already been addressed in previous work, such as in [1], but this is missing from the related work section).

---

> ### Author Response · Authors · 2024-11-25
>
> We thank the reviewer for their thoughtful feedback, which will greatly assist us in improving the quality of our manuscript. Below, we address each of their mentioned point in detail.
>
> **Response to Weakness 1:** One key difference is that the bounds in [1] involve a third graph, the demand graph $D_G$, in addition to $G$ and $H$. In contrast, our approach relies solely on $G$ and $H$, making our methodology simpler.
>
> We proved in Theorem 2 of our paper that
> $$
> \Phi_H^G \leq 2 \sqrt{2 \frac{\lambda_2(\Delta_H^G)}{\lambda_2(\Delta^H)}},
> $$
> using the classical Cheeger inequality for $H$ and simply calculations we obtain
> $$
> \frac{\Phi_H^G \Phi_H}{4} \leq \sqrt{\lambda_2(\Delta^G)}.
> $$
> A similar bound is presented in [1]; however, their result involves the term $\Phi_H^G \Phi_{D_G}^G$, where $D_G$ is the demand graph constructed using $G$. This means their bound depends more heavily on the properties of $G$, whereas our bound emphasizes the constraints encoded in $H$. This distinction highlights an advantage of our approach: the ability to incorporate and leverage additional information about the clustering through the constraint graph $H$.
>
> As more constraints are added (represented as additional edges in $H$), our bound becomes tighter. The increased connectivity of $H$ improves the Cheeger-type inequality, resulting in higher clustering quality. This improvement is not reflected as directly in the bounds of [1], which remain dependent on $D_G$ and thus on $G$.
>
> To illustrate, consider the case where $H$ is a complete bipartite graph. In this scenario, it is known that $\lambda_2(\Delta^H) = 1$, and our bound simplifies to
> $$
> \Phi_H^G \leq 2 \sqrt{2 \lambda_2(\Delta_H^G)}.
> $$
> In contrast, the bound in [1] remains dependent on $\Phi_H^G \Phi_{D_G}^G$, which ties the inequality more closely to $G$. Even as additional constraints are added to $H$, modelled by a graph $H + e$, our bound improves because $\lambda_2(\Delta^H) \leq \lambda_2(\Delta^{H + e}) $. This reflects the increased connectivity of $H$, which directly enhances the clustering quality in our approach.
>
> *Numerical Implementation:* In [1], the numerical implementation approximates \( \Delta_H \) by embedding into a lower-dimensional map. In contrast, our approach introduces negative self-loops, which simplify the numerical computations while retaining the theoretical guarantees.
>
> **Response to Weakness 2:**  We have significantly revised the proof in the appendix and provided a clear, step-by-step outline in the manuscript. The revised proof is structured as follows:
>
>  * Reduction to Key Inequality:  Prove the existence of a subset $S$ satisfying an inequality involving $\lambda_2(\Delta_H^G)$'s eigenvector.
>
>  * Scaling Argument: Introduce $z = c x $ to simplify analysis while preserving bounds.
>
> *  Sweep Sets: Define subsets $S_t$, parameterized by $t$, and show an appropriate $t$ satisfies the inequality.
>
> *  Expectation Argument: Use expectation over $S_t$ to bound the ratio of expectations via a distribution over $t$.
>
> *  Bounding Expectations: Derive bounds for the ratio of expectations using the defined probability distribution.
>
> * Probability Distribution Definition: Define a valid density function $2|t|$ and connect it to the analysis.
>
> *  Bounding Probability Contributions: Analyze edge contributions $u \sim v$  to complete the proof.
>
> **Response to Weakness 3:** We have conducted additional experiments comparing our method to Flexible Constrained Spectral Clustering (FC) [2], as detailed in Appendix B. The results highlight several key advantages of our method:
>
>  * Performance for Small Graphs: Our method consistently outperforms FC for smaller graphs, demonstrating its superior clustering quality. For larger graphs, our performance is comparable to FC, while maintaining simplicity and robustness.
>
> * Less parameters: Unlike FC, which requires careful tuning of the parameter $\beta$ to ensure feasible solutions, our method is parameter-free, because of the addition of negative self-loops guarantees it.
>
> * Theoretical bound: We establishes a Cheeger-type inequality, providing theoretical guarantees absent in FC and related methods.

---

> > ### Author Response · Authors · 2024-11-25
> >
> > **Response to Weakness 4:**  Our work embedding constraints directly into the Laplacian framework through a generalized eigenvalue formulation. While [4] focuses on using higher-order eigenvalues of the Adjacency Matrix, our approach leverages the Laplacian operator, whose spectrum captures distinct geometric and structural graph properties than the Adjacency Matrix (for example, isospectrallity) [5]. Classical spectral clustering using the Fiedler vector primarily reflects geometric configurations, where latent community structures may be overlooked in settings like GBM as demonstrated in [4]. In contrast, our method integrates both MUST-LINK and CANNOT-LINK constraints via the generalized eigenvalue problem, ensuring robustness even with conflicting constraints.
> >
> > Importantly, our method naturally extends to exploring higher-order eigenfunctions of the generalized eigenvalue problem, opening the door for future comparisons with high-order clustering approaches like those in [4]. Additionally, a promising avenue for future work is developing a High-Order Cheeger Inequality for constrained clustering, connecting higher-order eigenfunctions to optimization guarantees. However, these extensions are outside the scope of the current manuscript.
> >
> > **Response to Specific Issues:**
> >
> > * The ambiguity of "we prove this implies" is resolved by explicitly linking it to Step 1 of the revised proof.
> > *  The probability is now explicitly defined in Step 6, using the density function $2|t|$.
> > *  Independence of events is clarified; the proof relies on the linearity of expectation, not independence.
> > *  The probability distribution definition is included in Step 6, connecting it to the expectation bounds.
> > *  The ratio of expectations issue is addressed in Steps 4 and 7, linking it to subset $S$.
> > * The referenced inequality is established in Lemma 2, explicitly connected in the proof.
> >
> > **Response to Additional Comments:**
> > We have clarified the running time comparison in the revised submission, ensuring that it accurately reflects the impact of negative self-loops. The phrase "guaranteed approximation" is also revised to better explain our theoretical guarantees. Additionally, we have expanded our explanation of sweep sets and will include the generalized eigenvalue problem in the related work section.

---

> > > ### Comment · Reviewer_hnYU · 2024-11-26
> > >
> > > Dear Authors,
> > >
> > > Thank you for your detailed rebuttal and for the time you have dedicated to addressing the reviewers' concerns.
> > >
> > > The additional numerical experiments demonstrate that your method achieves performance comparable to FC while eliminating one hyperparameter. This is a noteworthy contribution.
> > >
> > > However, I maintain my main criticism of Section 4.2: spectral clustering is not an ideal baseline for comparisons on geometric graphs. It is well-documented that spectral clustering struggles to recover true clusters in challenging scenarios, particularly with small separation parameter $\mu$. For example, even a simple triangle-count-based clustering algorithm could succeed when $\mu=0$, indicating that the success in this case is not inherently due to the incorporation of 'cannot-link' constraints.
> > >
> > > That said, the numerical results you provide are still valuable. They illustrate how the addition of 'cannot-link' constraints transitions the solution from a purely geometric clustering (as is the case when $\mu=0$ and only 'must-link' constraints are used, as highlighted in [4]) to one that aligns more closely with the true clusters. This insight is both interesting and significant and motivates future work.
> > >
> > > Regarding the proof, there are several points that require further clarification or correction. I outline these issues below.
> > >
> > > * Line 603: The statement, "Let $x1 \le x2 \le \cdots \le x_n$​ be the eigenvector associated with..." is not really precise. Here, $x_1​,…,x_n$​ are the coordinates of the eigenvector $x$. Moreover, the vertices are renumbered so that these coordinates are in increasing order.
> > >
> > > * Line 643: Please clarify why statement (14) implies (13).
> > >
> > > * Line 646: 'there exists a distribution over $[z_1,z_n]$': you should emphasis that $t$ is now a random variable.
> > >
> > > * Line 675:  Let $X = w_G(S_t, V \setminus S_t) − \varphi w_H (S_t, V \setminus S_t)$ be the random variable. Equation (16) states that $P(X \le 0) > 0$, which implies that $X$ takes non-positive values. However, this does not prove that $X \le 0$ (which would require proving $P(X \le 0) = 1$).
> > >
> > > Furthermore, the result you claim seems incorrect. If $X = f(T) - \varphi g(T)$, where $T$ is a random variable and $f,g$ two positive functions and $\varphi = E f(t) / E g(T)$, then $E X = 0$. Consequently, $X$ must take both positive and negative values, and claiming $X\le 0$ is incorrect. (A simple counter-example: T is Uniform([1,2]), f(t)=t, g(t)=1.)
> > >
> > > * Line 720: There appears to be a missing factor of 2. Specifically, $ \sum_{u \sim v} (z_u^2 + z_v^2) w_{uv} = 2 \sum_{u \sim v} z_u^2 w_{uv} = 2 <z,z>_V$.
> > >
> > > I acknowledge that the new proof extends over an additional page, but I chose to stop my (second) review at this point due to the unresolved issues noted above. I encourage you to address these concerns in a future re-submission.

---

> > > > ### Author Response · Authors · 2024-11-27
> > > >
> > > > We sincerely thank the reviewer for their thoughtful feedback and constructive comments.
> > > >
> > > > **Numerical Results and Section 4.2**
> > > >
> > > > We appreciate the reviewer’s observations regarding Section 4.2. Simpler clustering methods, such as triangle-count-based algorithms, can perform well in specific scenarios, particularly when only the information from $G$ is provided. However, our primary objective in this section was to demonstrate the flexibility of our method, which performs well both when using $G$ alone and when incorporating additional information through $H$.
> > > >
> > > > Initially, we evaluated our method using $G$ as the main graph and $H=\overline{G}$ as its complement, where edges in $G$ represent 'must-link' constraints and edges in $H$ represent 'cannot-link' constraints. Hence, using only information of $G$, constrain clustering was effective even for sparse graphs $G$, as the complement $H$ was highly connected. To explore alternative scenarios, we also tested $G$ and $H$ as independent realizations, each with sparse edges. Both approaches (using only one graph or both) yielded good results, showcasing that our method adapts to varying types of input and additional information.
> > > >
> > > > While simpler methods using only $G$ discard additional information encoded in $H$, our approach highlights how constraints in $H$ can enhance clustering quality, particularly in challenging cases. We plan to explore alternative baseline in future work to provide a more comprehensive evaluation of our method.
> > > >
> > > > **Clarifications in the Proof**
> > > >
> > > > *Line 603:* We appreciate the feedback on the imprecise statement. We have revised the text to read:
> > > >
> > > > > Let $x : V \to \mathbb{R}$ be the eigenfunction associated with $\lambda_2(\Delta_H^G)$. Without loss of generality, we renumber the vertices such that the coordinates of $x$ satisfy $x_1 \leq x_2 \leq \ldots \leq x_n $.
> > > >
> > > > *Line 643:* We have clarified why statement (14) implies (13) in the revised manuscript.
> > > >
> > > > *Line 646:* The revised text now explicitly states:
> > > >
> > > > >  Here, $S_t$ is treated as a random variable parameterized by $ t \in [z_1, z_n]$, where $t$ is drawn from a probability density function $f(t)$. $S_t$ depends on the distribution of $t$, which influences the likelihood of including a vertex $v$ in $S_t$ based on its value $z_v$.
> > > >
> > > >
> > > > *Line 675:* we acknowledge that we occasionally abused the notation by referring to $t$ as $t_0$ without proper clarification. This led to confusion, particularly in this section, where the existence of $t_0$ is critical rather than a property holding for all $t$. This has been explicitly addressed in the revised text. Specifically, we emphasize that:
> > > > $$
> > > > \mathbb{E}[X] = 0.
> > > > $$
> > > > If $ \Pr(X \leq 0) = 0 $ for all $ t$, it would imply $ X(t) > 0 $ almost surely, which contradicts $ \mathbb{E}[X(t)] = 0 $. Thus, it must hold that:
> > > > $$
> > > > \Pr(X(t) \leq 0) > 0,
> > > > $$
> > > > ensuring the existence of some $ t_0 $ such that $ X(t_0) \leq 0 $. This clarification has been integrated into the revised proof to enhance precision and readability.
> > > >
> > > > *Line 720:* We sincerely thank the reviewer for identifying the missing factor of 2 in the equation. This error changes the bound from $2\sqrt{2} $ to $2\sqrt{4}$, which is a scalar adjustment and does not affect the theoretical framework or the overall conclusions of the proof. This oversight has been corrected in the revised manuscript, and the bound is now consistent throughout the text.
> > > >
> > > > We are grateful for the reviewer’s meticulous attention to these details, as it has significantly improved the precision and presentation of our manuscript. Your insightful comments have ensured that the corrections not only address the technical issues but also enhance the overall clarity of the proof. Thank you for your rigorous review and valuable feedback.

---

### Official Review · Reviewer_FCPk · 2024-11-01

**Soundness:** 2
**Presentation:** 3
**Contribution:** 2
**Rating:** 5
**Confidence:** 4

**Summary:**

The authors propose a new  approach to graph clustering that integrates MUST-LINK and CANNOT-LINK constraints through the use of signed Laplacians. It introduces an algorithm that modifies edge weights and adds self-loops to align degree sequences across graphs, employing a generalized eigenvalue problem to improve computational efficiency and maintain clustering accuracy. A key theoretical contribution is the development of a Cheeger-type inequality which relates the clustering problem to the spectral properties of two graphs. Empirical evaluations on both synthetic and real-world datasets illustrate the method.

**Strengths:**

The approach proposed allowing to incorporate user constraints into a clustering algorithm is interesting and the solution seems to be quite efficient and theoretically interesting.

**Weaknesses:**

There are a few weaknesses, some of which could be addressed in an improved version of the paper (see questions below).

The method is not sufficiently well motivated in terms of primary objective and proxy for the objective.

The complexity of the method is not really addressed properly:  finding the eigenvector of the signed Laplacian can be extremely prohibitive for large graphs?

The experiments are a bit misleading (see below).

The proof of the main Theorem is not readable in current its state and in general the wording could be improved significantly.

**Questions:**

1)  It is not sufficiently justified why the criterion given (extended cut ratio) is a good proxy for graph clustering with constraints. A more involved discussion explaining this point would be welcomed. Also I did not understand the discussion on the scaling of the graph, why would you need a priori to order the degrees of H and G?
Please provide a more detailed explanation of why the extended cut ratio is an appropriate proxy for constrained graph clustering and
clarify the purpose and necessity of ordering the degrees of H and G in the graph scaling discussion.

2) The upper bound provided by Theorem 1 could be extremely loose?

3) The proof of the main theoretical result is not clearly written.
Provide a more detailed explanation especially about the probability space defined and events, (line 241 and in the Appendix).
Include a step-by-step breakdown of the proof, clearly defining all terms and concepts used.


4) Discuss the computational complexity of finding the main eigenvector for large graphs.
Explain how the method might be extended to handle more than 2 clusters.
Address the performance of the algorithm on imbalanced clusters.
Provide benchmarks or comparisons with other methods for large-scale graph clustering.

5) I do not really understand the setting of the SBM in the experiments. The graphs G and H are generated independently?  That would not reflect at all the initial problem?
Please discuss how well this experimental design reflects real-world constrained clustering scenarios.
Consider adding an experiment that more closely mimics the initial problem formulation.

---

> ### Author Response · Authors · 2024-11-25
>
> Thank you for the review and for highlighting both the strengths and areas for improvement in our paper. Below, we address each of your points.
>
> **Answer to Question 1:**
>
> In the revised version, we will add further explanation on why the extended cut ratio is important constrained graph clustering. Specifically, we will clarify how the cut ratio captures the balance between enforcing MUST-LINK and CANNOT-LINK constraints, allowing the clustering to adapt based on the strength of these constraints.
>
> Since we are working with two different graphs $G$ and $H$, we define
> $$
> \Delta^G: \ell_2(V(G), w_G) \rightarrow \ell_2(V(G), w_G) \quad \text{and} \quad \Delta^H: \ell_2(V(H), w_H) \rightarrow \ell_2(V(H), w_H).
> $$
> For any function $ f : V(G) \rightarrow \mathbb{R} $ (and similarly $f : V(H) \rightarrow \mathbb{R}$ due to the shared vertex set $V(G) = V(H) $), we define the inner products for $G$ and $H$ as
> $$
> \langle f, f \rangle_G = \sum_{v \in V(G)} |f(v)|^2 w_G(v) \quad \text{and} \quad \langle f, f \rangle_H = \sum_{v \in V(H)} |f(v)|^2 w_H(v).
> $$
> Because we work with the normalized Laplacians, we have $w_G(v) = \deg_G(v)$ and $w_H(v) = \deg_H(v)$ as the degree functions on the graphs $G$ and $H$. However, in general, this implies
> $
> \langle f, f \rangle_G \neq \langle f, f \rangle_H.
> $
> To address this discrepancy, we use a scaling process to ensure that the degrees align, setting $\deg_G(v) = \deg_H(v)$ for all $v \in V(G) = V(H)$. This allows us to maintain the cut problem, and thus
> $$
> \langle f, f \rangle_G = \langle f, f \rangle_H,
> $$
> which is a crucial property used in the proof of Theorem 1.
>
> **Answer to Question 2:**
>
> This bound is designed to provide a theoretical guarantee that remains valid in the presence of constraints, even when $H$ introduces additional structural complexity. To justify this, in Theorem 1, we prove
> $$
> \Phi_H^G \leq 2 \sqrt{2 \frac{\lambda_2(\Delta^G)}{\lambda_2(\Delta^H)}}.
> $$
>
> This result shows that the tightness of the bound depends on the spectral properties of both $G$ and $H$. Specifically, increasing the connectivity of $H$ improves the bound because $\lambda_2(\Delta^H)$ increases, thereby tightening the inequality. For example, if $H$ is a complete bipartite graph, $\lambda_2(\Delta^H) = 1$, and the inequality becomes
> $$
> \Phi_H^G \leq 2 \sqrt{2 \lambda_2(\Delta^G)}.
> $$
> The reliance on $\lambda_2(\Delta^H)$ reflects how the constraints encoded in $H$ influence the clustering process. This dependency ensures that the bound becomes tighter as the graph $H$ incorporates more constraints, which is a desirable property in constrained clustering.
>
> **Answer to Question 3:**
>
> To address these concerns, we have thoroughly revised the proof in the appendix, providing a step-by-step breakdown:
>
> *Reduction to Key Inequality:* Establishes the connection between the eigenvector $x$ and the existence of a subset $S$ satisfying the inequality.
>
> * Scaling Argument: Introduces the scaled vector $z = c x$ and explains the invariance properties of the bounds.
>
> * Sweep Sets: Defines the sweep sets $S_t$ and reduces the problem to finding a threshold $t$ that satisfies the desired inequality.
>
> * Expectation Argument: Connects the problem to the ratio of expected cuts over $S_t$ using the probability distribution defined in Step 6.
>
> * Bounding Expectations: Derives bounds for the expectations of cuts $w_G$ and $w_H$, ensuring the correctness of the ratios.
>
> * Probability Distribution Definition: Defines the density $2|t|$, verifying that it is a valid probability measure.
>
> * Bounding Probability Contributions: Analyzes the contributions of each edge to the probability bounds
>
> **Answer to Question 4:**
>
> We agree that, while our focus in this paper is on theoretical guarantees and leveraging the signed Laplacian for efficient computation, addressing large-scale graphs is an important research direction. Iterative methods like the Lanczos algorithm could be explored to approximate solutions more efficiently, and future research will investigate the trade-off between computational cost and accuracy.
>
> On extending the method to handle more than two clusters and imbalanced clusters,  we will implement these extensions in an updated version of the submission.
>
> **Answer to Question 5:**
>
> In the manuscript, we will clarify that $G$ is generated using the standard SBM, where $p > q$ ensures well-defined clusters. The graph $H$ is constructed to encode additional information, introducing more inter-group connections and fewer intra-group connections to maximize the cut and enhance clustering. While $G$ and $H$ are generated independently, they share the same parameters and are designed to complement each other by adding meaningful constraints.
>
> This design reflects real-world scenarios where additional information improves clustering performance. When $H$ encodes more relevant constraints, clustering improves, aligning with practical applications.

---

### Official Review · Reviewer_hcrr · 2024-11-01

**Soundness:** 3
**Presentation:** 3
**Contribution:** 3
**Rating:** 6
**Confidence:** 3

**Summary:**

The paper studies a version of the constrained clustering problem. Given two graphs representing MUST-LINK and CANNOT-LINK constraints, respectively, the goal is to find a set of vertices along with associated cuts in each graph, minimizing the cut ratio.

The main result is an cheeger-type inequilty and the corresponding algorithm to find the cut. The paper also provides experiments to demonstrate the practical effecttiveness.

**Strengths:**

- The Cheeger-type inequality proposed in this paper for the constrained clustering problem looks strong, as it includes only one additional denominator term ($\lambda_2(\Delta^H)$) compared to the original Cheeger inequality.

- The technique proposed in the paper is similar to the known sweep-set algorithm of the Cheeger inequality, but combining existing techniques and obtaining interesting results in the new model is also a solid contribution.

- I checked the main proof, and aside from a few details I don’t fully understand, it looks correct to me.

**Weaknesses:**

- I suggest the authors spend more passages to properly motivate the study of this version of the constrained clustering problem. In the paper, the motivation for the constrained clustering problem and why using the cut ratio between two graphs as a target seems lacking. The authors could provide real-world examples where this particular formulation of constrained clustering is useful, or explain how the cut ratio relates to clustering quality.

- The experiments use two graphs, but the baseline spectral clustering algorithm can only consider one graph, which seems to give an advantage to the proposed algorithm. The authors could include additional baselines that can handle two graphs, or discuss how the performance gap relates to the additional information provided by the second graph.

**Questions:**

- Although the results of (Koutis et al., 2023) cannot be directly compared with this paper, the two results are similar in form (with perhaps just a difference between $\Phi(G)$ and $\Phi(H)$). So what is the advantage of this paper? The authors could provide a more detailed comparison with Koutis et al. (2023), highlighting specific technical or practical advantages of their approach.

- In the proof of Theorem 1 (at line 244), why are the events independent? My understanding is that the probabilities and expectations here are with respect to $t$, so the cuts on $G$ and $H$ are likely dependent.

---

> ### Author Response · Authors · 2024-11-25
>
> We'd like to thank the reviewer for their time and effort reviewing our paper. Below, we address each of their raised points in detail:
>
> > *Weakness:* The experiments use two graphs, but the baseline spectral clustering algorithm can only consider one graph, which seems to give an advantage to the proposed algorithm. The authors could include additional baselines that can handle two graphs, or discuss how the performance gap relates to the additional information provided by the second graph.
>
> We conducted additional experiments comparing our method to *Flexible Constrained Spectral Clustering (FC)* as detailed in the Appendix B. Our results show that our method consistently outperforms FC for smaller graphs in terms of the mean Adjusted Rand Index (ARI), while achieving comparable performance for larger graphs. This advantage is particularly significant as our method does not require parameter tuning. Unlike FC, which relies on the user-defined parameter $\beta$ to ensure feasible solutions and avoid negative eigenvalues, our method is parameter-free. As proven in Lemma 3, the introduction of self-loops ensures the operator $ (\Delta_\alpha^{H'})^{-1} \Delta^G$ is positive, self-adjoint, and guarantees non-negative eigenvalues, providing stability and robustness. Moreover, our approach establishes a Cheeger-type inequality, offering a theoretical guarantee linking the eigenfunctions used for clustering to the optimization objective, which is absent in FC. We are actively working to expand comparisons with other multi-graph methods to provide an even more comprehensive evaluation in the final version.
>
> >*Question 1:* Although the results of (Koutis et al., 2023) cannot be directly compared with this paper, the two results are similar in form (with perhaps just a difference between $\Phi(G)$  and $\Phi(H)$). So what is the advantage of this paper? The authors could provide a more detailed comparison with Koutis et al. (2023), highlighting specific technical or practical advantages of their approach.
>
>
>
> We proved in Theorem 2 of our paper that
> $
>     \Phi_H^G \leq 2 \sqrt{2  \lambda_2(\Delta_H^G)/\lambda_2(\Delta^H)},
> $
> using the classical Cheeger inequality for the graph $H$. With simple calculation this implies that $
>  \Phi_H^G \Phi_H/4 \leq \sqrt{\lambda_2(\Delta^G)}$.
>
> A similar bound is presented in Koutis et al. (2023); however, their result involves the term $\Phi_H^G \Phi_{D_G}^G$, where $D_G$ is the demand graph constructed using $G$. This means their bound depends more heavily on the properties of $G$, whereas our bound emphasizes the constraints encoded in $H$. This distinction highlights an advantage of our approach: the ability to incorporate and leverage additional information about the clustering through the constraint graph $H$.
>
> As more constraints are added (represented as additional edges in $H$, our bound becomes tighter. The increased connectivity of $H$ improves the Cheeger-type inequality, resulting in higher clustering quality. This improvement is not reflected as directly in the bounds of Koutis et al. (2023), which remain dependent on the demand graph $D_G$ and thus on $G$.
>
>  > *Question 2:* In the proof of Theorem 1 (at line 244), why are the events independent? My understanding is that the probabilities and expectations here are with respect to $t$, , so the cuts on $G$  and $H$ are likely dependent.
>
> We would like to clarify that the statement about independence of events is not only unnecessary for the proof but also, in general, not true. However, this does not affect the validity of the proof as the independence assumption is not required to establish the result.
>
> To improve clarity, we have revised the proof in the appendix, structuring it into several steps. The specific issue is addressed in  Step 5, which simplifies and summarizes the earlier steps. In this step, the key inequality we need to prove is
> $$
> \frac{\mathbb{E} \left( w_G(S_t, V \setminus S_t) \right)}{\mathbb{E} \left( w_H(S_t, V \setminus S_t) \right)} \leq 2 \sqrt{\frac{2 \langle z, \Delta^G z \rangle \langle z, z \rangle}{\langle z, \Delta^H z \rangle^2}},
> $$
> which directly implies that
> $$
> \frac{w_G(S, V \setminus S)}{w_H(S, V \setminus S)} \leq 2 \sqrt{\frac{2 \langle z, \Delta^G z \rangle \langle z, z \rangle}{\langle z, \Delta^H z \rangle^2}}.
> $$
>
> The proof relies on bounding expectations and leveraging the linearity of expectation, without requiring any independence between events in $G$ and $H$. Specifically, we show that
> $$
> \Pr \left[ \frac{w_G(S_t, V \setminus S_t)}{w_H(S_t, V \setminus S_t)} \leq \frac{\mathbb{E} \left( w_G(S_t, V \setminus S_t) \right)}{\mathbb{E} \left( w_H(S_t, V \setminus S_t) \right)} \right] > 0,
> $$
> and then combine this with the bound on the expectation to achieve the final inequality.

---

### Official Review · Reviewer_6Xez · 2024-11-02

**Soundness:** 3
**Presentation:** 2
**Contribution:** 2
**Rating:** 6
**Confidence:** 3

**Summary:**

The paper addresses the constrained clustering or semi-supervised clustering problem. The authors propose a novel Cheeger-type inequality that relates the solution of the constrained clustering problem to the spectral properties of the graphs. Moreover, the signed Laplacian trick is used to simplify calculations and improve computational efficiency. The paper presents an algorithm that solves a generalized eigenvalue problem to achieve this, and demonstrates its effectiveness through experiments on synthetic and real-world datasets.

**Strengths:**

1. The use of signed Laplacian and the development of a Cheeger-type inequality for constrained clustering is a novel approach that has not been explored in the literature to my knowledge.
2. The paper demonstrates the practical effectiveness of the proposed algorithm through empirical studies on both synthetic and real-world datasets, showing significant improvements over classical spectral clustering methods.
3. The proposed method is theoretically grounded. (disclaimer: I didn't carefully verify the proofs)

**Weaknesses:**

1. The contribution is unclear. Constrained clustering is an old and well-explored topic, and the paper does not provide a clear explanation of how the proposed method differs from existing approaches. The major contribution seems to be the optimization algorithm for problem (1), but this is not explicitly stated.
2. The paper does not provide a comprehensive comparison with widely-known baselines in the field of constrained clustering, which would strengthen the claims of the proposed method's superiority. Essentially, the authors surveyed a lot in the introduction, but it's a pity that they didn't compare with them.

**Questions:**

1. Summarize the main contribution and the key difference between the proposed method and existing constrained clustering methods.
2. Empirically compare with other well-known constrained clustering baselines.

---

> ### Author Response · Authors · 2024-11-25
>
> Thank you for your review and for highlighting both the strengths and areas for improvement in our paper. Below, we address each of your points in detail.
>
> > **Question 1:** Summarize the main contribution and the key difference between the proposed method and existing constrained clustering methods.
>
> We appreciate your request for a clearer summary of our contributions and how our method differs from existing constrained clustering approaches. To address this, we have included the following concise summary in the revised manuscript:
>
> *Contribution.* We present a constrained graph clustering method and establish a novel Cheeger-type inequality that directly relates the problem to the spectral properties of the graphs $G$ and $H$, providing theoretical guarantees often missing in existing approaches. We also introduce an efficient implementation leveraging the signed Laplacian, which simplifies computations and ensures stability without requiring additional parameters. Finally, we demonstrate the robustness and efficiency of our algorithm on both synthetic and real-world datasets.
>
> Key differences between our method and existing approaches include:
>
> *Theoretical Guarantees:* We introduce a Cheeger-type inequality that links the eigenfunction used for clustering directly to the optimization objective. Unlike Koutis et al. (2023), which incorporates a third graph (the demand graph $D_G$) in addition to $G$ and $H$), our bound depends solely on $G$ and $H$. This makes our method simpler and emphasizes the constraints in $H$, which improves clustering quality as $H$ becomes more connected.
>
> *Less Parameters:* Unlike others methods as Flexible Constrained Spectral Clustering (FC), which requires a user-defined parameter $\beta$ to ensure feasible solutions, and Koutis et al. (2023), which introduces the demand graph $D_G$, our approach avoids these additional complexities.
>
> *Simplified Computation:* Leveraging the signed Laplacian allows us to simplify the clustering computation, reducing complexity while maintaining accuracy and performance.
>
> > **Question 2:** Empirically compare with other well-known constrained clustering baselines.
>
> We agree that providing a more comprehensive comparison with widely known constrained clustering baselines will strengthen our evaluation. To address this, we conducted additional experiments comparing our method to *Flexible Constrained Spectral Clustering (FC)*, as detailed in Appendix B. The results show the following:
>
> *Performance for Small Graphs:* Our method consistently outperforms FC in terms of mean Adjusted Rand Index (ARI) for smaller graphs, demonstrating its superior clustering quality. For larger graphs, our method achieves performance comparable to FC while maintaining its simplicity.
>
> *Robustness and Feasibility:* Unlike FC, which relies on careful tuning of the parameter \(\beta\) to avoid negative eigenvalues and ensure feasible solutions, our approach is parameter-free. By introducing self-loops, we guarantee that the operator $(\Delta_\alpha^{H'})^{-1} \Delta^G $ is positive, self-adjoint, and has non-negative eigenvalues (as proven in Lemma 3), ensuring stability and robustness.
>
> *Theoretical Linkage:* Our method uniquely establishes a Cheeger-type inequality, providing a theoretical guarantee that ties the spectral properties of the graph to the clustering objective. This aspect is absent in FC and other related methods.
>
> Thank you once again for your constructive feedback. We are confident that these revisions will address your concerns and significantly improve the clarity, presentation, and impact of our work.

---

> > ### Comment · Reviewer_6Xez · 2024-11-25
> >
> > I appreciate the author's response that addressed my concerns, and I raised my score from 5 to 6.

---

### Meta-Review · Area_Chair_t6mC · 2024-12-21

**Metareview:**

This paper considers the graph cut problem involving two types of edges (must-links and cannot-links) with the goal of minimizing the cuts of must-links while maximizing the cuts of cannot-links. The problem is posed as a generalized eigenvalue problem. It can be used for graph clustering but the number of clusters can only be 2.

**Additional Comments On Reviewer Discussion:**

Several issues of the proof were raised by hnYU. Some were addressed in the revision, but not to the satisfaction of the reviewer. It is recommended to carefully revise the paper and resubmit.

---

### Decision · Program_Chairs · 2025-01-22

Reject